# Navigating Nepal's health financing system: A road to universal health coverage amid epidemiological and demographic transitions

Resham B. Khatri[1,2]*, Pratik Khanal[3], Dipendra Singh Thakuri[2,4], Prabesh Ghimire[5], Mihajlo Jakovljevic[6,7,8]

1 Health Social Science and Development Research Institute, Kathmandu, Nepal, 2 School of Public Health, Faculty of Medicine, University of Queensland, Brisbane, Australia, 3 Bergen Centre for Ethics and Priority Setting in Health (BCEPS), Department of Global Public Health and Primary Care, University of Bergen, Bergen, Norway, 4 Poche Centre for Indigenous Health, University of Queensland, Brisbane, Australia, 5 Tribhuwan University, Institute of Medicine, Kathmandu, Nepal, 6 UNESCO-TWAS, The World Academy of Sciences, Trieste, Italy, 7 Shaanxi University of Technology, Hanzhong, China, 8 Department of Global Health Economics and Policy, University of Kragujevac, Kragujevac, Serbia

* rkchettri@gmail.com

## Abstract

### Background

Nepal has been undergoing demographic and epidemiological transitions, marked by an increasing burden of non-communicable diseases (NCDs) and injuries. These transitions have led to financial implications, including rising out-of-pocket (OOP) expenses. This study reviews and synthesizes evidence on the status, issues and challenges in health financing system, policies, and programs to achieve universal health coverage (UHC) in Nepal.

### Methods

We conducted a scoping review of literature on Nepal's health financing system, policies, and programs. A search strategy was developed using keywords related to two core concepts: health financing and universal health coverage. Grey literature was identified from the web pages of relevant ministries and organizations. A total of 148 studies/policy documents published in Nepali and English up to 31 December 2024 were included. Policies and content related to the health financing system were reviewed to understand the status, issues and challenges of health financing functions, and UHC . A framework-guided deductive content analysis approach was employed, and findings were interpreted using the three UHC components: service coverage, population coverage, and financial coverage.

**Data availability statement:** All relevant data are within the manuscript.

**Funding:** The author(s) received no specific funding for this work.

**Competing interests:** The authors have declared that no competing interests exist.

**Abbreviations:** BHS, Basic Health Services; CHEs, Current Health Expenditure; CMNN, Communicable, Maternal, Neonatal, And Nutritional Diseases; COPD, Chronic Obstructive Pulmonary Disease; CVDs, Cardiovascular Diseases; EPF, Employee Provident Fund; FY, Fiscal Year; GDP, Gross Domestic Product; GoN, Government of Nepal; HIB, Health Insurance Board; IHD, Ischemic Heart Disease; LMICs, Lower Middle-Income Countries; MCH, Maternal and Child Health Services; MOHP, Ministry of Health and Population; NCDs, Non-Communicable Diseases; NHA, National Health Account; NHF, National Health Financing; NHIP, National Health Insurance Program; NHS, Nepal Health Sector; NPR, Nepalese rupees; OOP, Out-of-Pocket; PEN, Package of Essential NCD; PHC, Primary Health Care Centers; PHEs, Public health emergencies; PHS, Public Health Service; RTAs, Road Traffic Accidents; SES, Socio-Economic Status; SHSDB, Social Health Security Development Board; SSF, Social Security Fund; SSS, Social Security Scheme; SSU, Social Service Unit; UHC, Universal Health Coverage.

## Results

Nepal's health policy documents prioritize financial protection for low-income people and target groups through social health protection programs/schemes. However, multiple social health protection schemes coexist with fragmented risk pooling and low efficiency in health financing. OOP expenditure is high at 54.2%, with 10% of the population facing catastrophic health expenditures. Injuries and chronic morbidities contribute significantly to this burden, with 70% of injury-related and 62% of NCD-related expenses borne through OOP payments. Despite efforts to improve financial risk protection, the National Health Insurance Program (NHIP) suffers from low population coverage (28%), low renewal rate (54%), and financial sustainability issues (as provider payments exceed revenue collection). The UHC service coverage index, though improving, was only 54 out of 100 in 2021 reflecting limited health system capacity and insufficient readiness to address health challenges, including those posed by shifting demographics and the growing burden of NCDs. Nepal's total health expenditure remains around 2% of GDP, with persistent inefficiencies in resource allocation, fiscal decentralization, and budget absorption.

## Conclusions

Nepal's health financing policies align with UHC goals, yet critical gaps remain in multiple dimensions . Issues such as inefficiencies, underfunding, and fragmented social health protection schemes limit equitable access to quality health care. Therefore, comprehensive structural reforms-spanning legal, institutional, and policy frameworks-are urgently needed. Key reforms include: (1) merging or harmonizing existing social health protection schemes for efficient pooling and purchasing; (2) enhancing domestic health financing through increased health funding (≥5% of GDP) via payroll contributions, progressive taxation, and earmarked sin taxes; (3) reforming NHIP to mandatory enrollment starting from formal sector, subsidizing premium for informal sector and free coverage for disadvantaged groups, alongside strengthening policy implementation including accrediting of health facilities, ensuring service quality, prioritising and expanding coverage packages with strategic purchasing from all public and private health facilities; and (4) equitable public financing to ensure needs-based allocation across government levels that respond to demographic and epidemiological patterns. Further research is needed to assess hybrid tax and premium based insurance models, strategic purchasing optimization, and digital health innovations for financial sustainability.

## Introduction

Nepal's healthcare delivery system comprises a mixed network of public and private health facilities (HFs). Public HFs operate across federal, provincial, and community levels, providing Basic Health Services (BHS) and specialized care. The

BHS—essential primary care is delivered through basic health service centers, health posts, primary healthcare centers, and basic hospitals [1], while specialized care is delivered via referral hospitals (e.g., provincial, federal, teaching, and super-specialty hospitals). Private HFs, concentrated in urban areas, offer services ranging from routine care to advanced treatments through hospitals, clinics, and pharmacies [2].

In line with its healthcare delivery system, Nepal employs a mixed health financing system, combining publicly funded health programs and services with private payment for healthcare services [3, 4]. The Government of Nepal (GoN) allocates funds to the health sector through its budget, pooling resources from various financing sources to reimburse healthcare providers [3]. Major sources include public funding (e.g., direct and indirect taxes, customs duties), private contributions (e.g., household out-of-pocket (OOP) payments, private insurers), and grants or loans from external development partners [4].

Over the past three decades, Nepal has witnessed significant changes in its demographic and epidemiological disease patterns [5]. These changes include general population growth (from 18.4 million in 1991 to 29.9 million in 2021) and an increase in life expectancy (from 54.9 to 71.2 years), leading to a rise in the elderly population (aged >60 years) from 5.8% to 10.21% during the same period [6]. There has also been a shift in the disease burden, transitioning from infectious diseases to non-communicable diseases (NCDs) and other conditions such as unplanned urbanization, climate change, antimicrobial resistance, and injuries [7]. For instance, the proportion of the population reporting NCDs increased from 6.5% in 1995/96 to 18.5% in 2022/23 [8].

In 2019, out of an estimated 193,331 deaths in Nepal, 71.1% were attributed to NCDs, 21.1% to communicable, maternal, neonatal, and nutritional diseases (CMNN), and 7.8% to injuries [7]. Among NCDs, cardiovascular diseases (CVDs) were the leading cause of death, accounting for 24% of fatalities, with higher rates among males (26.8%) than females (20.7%). Chronic obstructive pulmonary disease (COPD) caused 16.3% of deaths, up from 6.1% in 1990, while ischemic heart disease (IHD) accounted for 12.3%, up from 4% [7]. Unplanned urbanization has significantly contributed to the disease burden, particularly in major urban cities compared to rural areas [9]. Additionally, other emerging challenges include the emergence of vector-borne diseases and increased threats from public health emergencies (PHEs), such as natural disasters, floods, suicide, and road traffic accidents (RTAs) [7]. These transitions have profound implications for morbidity, as measured by Disability-Adjusted Life Years (DALYs), and mortality, particularly in terms of Years of Life Lost (YLL) [7].

Nepal's epidemiological shifts and demographic transitions, combined with PHEs, could pose significant challenges to achieving Universal Health Coverage (UHC) by 2030. This is particularly true if financial barriers to accessing health care remain unaddressed. The lack of adequate financial risk pooling has led to high OOP expenditures, which have consistently accounted for 55% to 60% of current health expenditure (CHE) over the past two decades [10], which is significantly higher than the global average of 28.16%[11] and the WHO-recommended threshold of 15–20% to prevent financial hardship [12]. For high-income countries, the average OOP expenditure is 16.96%, while for Lower Middle-Income Countries (LMICs), it is 38.11%[12]. Comparatively, neighboring countries like Bangladesh report very high OOP expenditures at 72.99%, while India and Sri Lanka have lower OOP expenditures than Nepal (49.82% and 43.64%, respectively)[13]. In contrast, Thailand, with its stronger public health system, has reduced OOP expenditures to below 10% through robust public financing and risk-pooling mechanisms [11]. Similarly, Nepal's reliance on external funding for health (around 10.35% of total health expenditure) is slightly higher than the regional average of 9.89% for the WHO South East Asia Region, but lower than the LMICs average of 18.2%, underscoring Nepal's continued dependence on donor support [11].

Alarmingly, one in ten Nepalese households spend more than 10% of their income on health care, and 1.7% of the population falls into poverty annually due to OOP payments [14]. Despite growing awareness among stakeholders about the shifting disease burden and high OOP expenses, Nepal's health system remains inadequately prepared to address these challenges. The system faces limitations in service capacity, resilience, and the ability to manage the health financing implications of the emerging burden of diseases and conditions [15–18]. This highlights the urgent need for Nepal to strengthen financial risk pooling mechanisms to reduce reliance on OOP payments and improve financial protection.

 

Nepal's health system and policy began evolving after the establishment of a constitutional monarchy in 1990 (replacing party-less Panchayat government) [19], followed by the introduction of the first national health policy in 1991[20]. Subsequent political turmoil and transitions, including the Maoist insurgency, and formation of political alliances, led to the adoption of the Interim Constitution in 2007 [21]. In 2025, Nepal adopted a new Constitution, establishing a federal republic and a federalized health system [22]. Grounded in socialist values, the constitution recognizes BHS as a fundamental right, mandating their provision free of charge to all citizens, along with additional health services through the national health insurance program (NHIP) [22]. In line with these constitutional commitments, several health policies have been introduced and implemented to ensure financial protection and improve service coverage. These include Nepal's Health Sector (NHS) Strategy (2023–2030), the National Health Financing (NHF) Strategy (2023–2033), and the Health Insurance (HI) Act 2017 [23–26]. In 2016, the GoN launched the NHIP to address financial barriers to health care and reduce OOP expenses [14]. However, the NHIP faces significant challenges, including low population coverage, high dropout rates, and an inadequate benefit package (in terms of services and cost coverage) [14, 27, 28]. To fulfil its constitutional mandate of providing health services, the GoN has implemented various social health protection programs and schemes, such as BHS, the Safe Delivery Incentive Program (Aama Program), the Free Newborn Care Program, the Impoverished Citizen Treatment Fund, special financial benefits for health treatment, and social service units, among others [3, 14]. Nepal has committed to achieving UHC and other health-related targets under Sustainable Development Goal Three (SDG 3) by 2030 [29]. Despite substantial investments, efforts, and progress, financial health protection initiatives have yet to translate into effective health coverage for all [14, 30]. Existing social protection programs and schemes are characterized by overlapping entitlements regarding population coverage and benefit packages, and fragmentation that limits risk pooling across population groups [30–32].

In Nepal, a large body of literature focuses on factors associated with health insurance, service utilization status [28, 33, 34], user satisfaction with health insurance schemes [35], household-level healthcare expenditure patterns [36], and causes of dropout from health insurance [37]. These studies are small-scale, standalone or community-level studies that examine community- or service-level drivers of financial health protection. There is a notable gap in comprehensive analyses of the health financing trajectory in relation to constitutional mandates, national policies, and health programs since the 1990s. Specifically, policy reviews that analyze the health financing landscape of Nepal using the three dimensions of the UHC (i.e., population, service, and financial coverage) and health financing functions remain scarce. This study thus provides a comprehensive policy-level analysis of Nepal's health financing trajectory using a UHC lens. By systematically examining the evolution of health financing policies since the 1990s, this study assesses their alignment with constitutional mandates, financial risk protection commitments, and equity considerations amid epidemiological and demographic shifts. Using a mixed-method policy review approach, it bridges critical evidence gaps by evaluating the effectiveness and sustainability of financing mechanisms in addressing financial vulnerabilities of the population. This study offers new insights into how different financing mechanisms interact to advance UHC, contributing to a broader understanding of the policy landscape and informing future reforms for equitable health financing in Nepal. These findings will support program managers and policy stakeholders in ensuring that health financing policies and programs remain responsive to the country's evolving demographic and epidemiological patterns.

## Methods

### Study design

We conducted a scoping review of published literature on Nepal's health financing-related policies and programs, examining their status, issues, and challenges in achieving UHC. The review followed the Preferred Reporting Items for Systematic Reviews and Meta-Analyses Extension for Scoping Reviews (PRISMA-ScR) Guideline [38] [Supplementary Information, S1 Table]. We adhered to the methodological framework of scoping reviews proposed by Arksey and O'Malley (2005)[39],

which was further refined by Levac and colleagues (2010) [40]. This framework includes six steps: (a) identifying research questions, (b) identifying relevant documents, (c) selecting documents, (d) extracting and charting data, (e) summarizing and reporting findings, and (f) engaging public and expert involvement. The scoping review design is well-suited for mapping broad evidence landscapes on this topic.

### Identifying research question

The following question guided the scoping review: What are the status, issues, and challenges of the health financing system in Nepal in relation to progress toward UHC, in terms of population and service coverage, as well as financial risk protection?

### Identifying relevant documents

We used search terms combining two concepts with a geographical focus on Nepal: health financing (e.g., social protection, health financing, health insurance) and universal health coverage (population, service, and financial risk protection). We searched PubMed, Embase, and Scopus [Supplementary Information, S2 Table] and reviewed references from selected papers to identify additional studies. The first four authors, who are familiar with Nepal's health system and academic/grey literature, identified grey literature through team discussions, professional networks, Google searches, and government websites. Additionally, we searched using similar search terms on Google and government websites. The grey literature included survey reports, policy documents (constitutions, acts, regulations, policies, plans, strategies), conference presentations, and news articles. We compiled a list of documents, discussed their relevance, and reached a consensus on inclusion. We also reviewed Nepal's country profiles and reports from the Institute for Health Metrics and Evaluation (IHME), the World Bank, and the WHO.

### Selection of documents

We searched for all relevant documents addressing health financing and UHC in Nepal [Supplementary Information, S3 Table]. Records were imported and managed using EndNote 20. The inclusion criteria were: (i) relevance to health financing/UHC themes and (ii) a detailed focus on Nepal. Screening was initially conducted based on the title and abstract by the first author and further assessed by the second author. This was followed by a full-text screening, initially performed by the first author and reviewed by the third and fourth authors. The data search was conducted during the first week of May 2024 (updated during the first week of January 2025). We included relevant literature written and published in Nepali or English up until 31 December 2024 (no starting date was applied in the search). Grey literature was included to emphasize its relevance to policy gaps. To minimize any bias from grey literature, we primarily relied on official government reports and published official documents, cross-validated with peer-reviewed studies where available. In cases of limited grey literature, we consulted three independent PhD candidates with expertise in Nepal's health and financing systems to ensure data appropriateness, accuracy, and to mitigate bias. This approach was discussed and agreed upon by the authors to enhance data reliability.

### Extraction and charting of data

We extracted health financing content from the selected documents. The extracted data were discussed among the authors for analysis and interpretation, aligned with the UHC framework [41,42]. The content was organized into a Word file under UHC framework components (population, service, and financial coverage) and reviewed collaboratively. The first author led the data extraction and analysis. When inconsistencies or ambiguities arose, such as variations in terminology, differences in reported figures, or misalignment with UHC dimensions, these were flagged and systematically discussed among the authors. Resolution involved cross-checking with primary documents, seeking consensus through predefined

inclusion criteria, and documenting decision rationales. Any disagreements were addressed through deliberation, reconsideration of the review questions, and alignment with the analytical framework, ensuring relevance to policy content related to health financing status, challenges, and programs/services aligned with the UHC framework.

### Summarizing and reporting the findings

To guide our analysis, we adopted the three interlinked components of the UHC framework for health financing: population, service, and financial coverage [41]. The UHC framework encompasses population coverage (ensuring all people have access to the full range of quality health services they need, when and where they need them), financial coverage (protecting individuals from financial hardship), and service coverage (encompassing the full continuum of essential health services) [42]. Population coverage refers to access to health services by population groups, while service coverage refers to the health services or interventions available to the population. Finally, financial risk protection or coverage protects individuals from financial hardship when seeking health care services. Data analysis was conducted in two phases. In the first phase, health financing-related policies and programs were reviewed. Using data from peer-reviewed studies and grey literature, the status, issues, and challenges of the policies were presented, and the coverage status of UHC for each program/scheme was also presented. In the second phase, we further explained the status and challenges of UHC coverage. For this, data were analyzed using a framework-guided deductive content analysis (extracting data and fitting them into the pre-identified components of the framework). Then, we grouped similar content across each component of the UHC framework, synthesized the findings, and provided a narrative explanation.

### Public and experts' involvement

The scoping review did not directly involve patients or the public. However, we consulted professionals working in the field to identify relevant documents, policies, and guidelines on health financing (see Acknowledgements). To validate and confirm our findings, we reviewed expert opinions from publicly available platforms, such as YouTube and live-recorded videos of presentations and talks at national conferences. Notable events included the 'National Summit of the Health and Population Scientists in Nepal', organized by the Nepal Health Research Council in 2022, 2023, and 2024 and the international conference on 'Resilient and Inclusive Social Protection: Investing in Human Capital Development', organized by the National Planning Commission in 2022. Additionally, we consulted with three PhD candidates specializing in health economics and health systems, who have expertise in health financing in Nepal, to review the manuscript and ensure factual accuracy. Since the study used secondary data, ethical approval from an institutional review board was not required.

## Results

A total of 148 documents were included in the analysis including government documents (40), reports/surveys (36), online databases (4), conference presentations/ panel discussions (4), newspaper reports (3), and peer-reviewed articles identified through hand search (8) [Fig 1] [see details in Supplementary Information, S3 Table}.

### Health financing landscape

Table 1 provides an overview of the evolution of health financing policies, highlighting the associated issues and challenges [see Supplementary information, S1 Fig]. The constitutional focus on health and health financing has evolved across the three constitutions since 1990: the Constitution of the Kingdom of Nepal (1990) framed health as a non-binding directive, reliant on state budgets and donors, with high OOP costs (~70%) [19], the Interim Constitution of Nepal 2007 (2007–2015) established health as a fundamental right, mandating free basic health services and introduced equity-focused financing [21], and the Constitution of Nepal 2015, which institutionalized federal health governance, free basic health services, and mandates ensuring for citizens are not deprived of emergency care, and decentralized resource allocation [22].

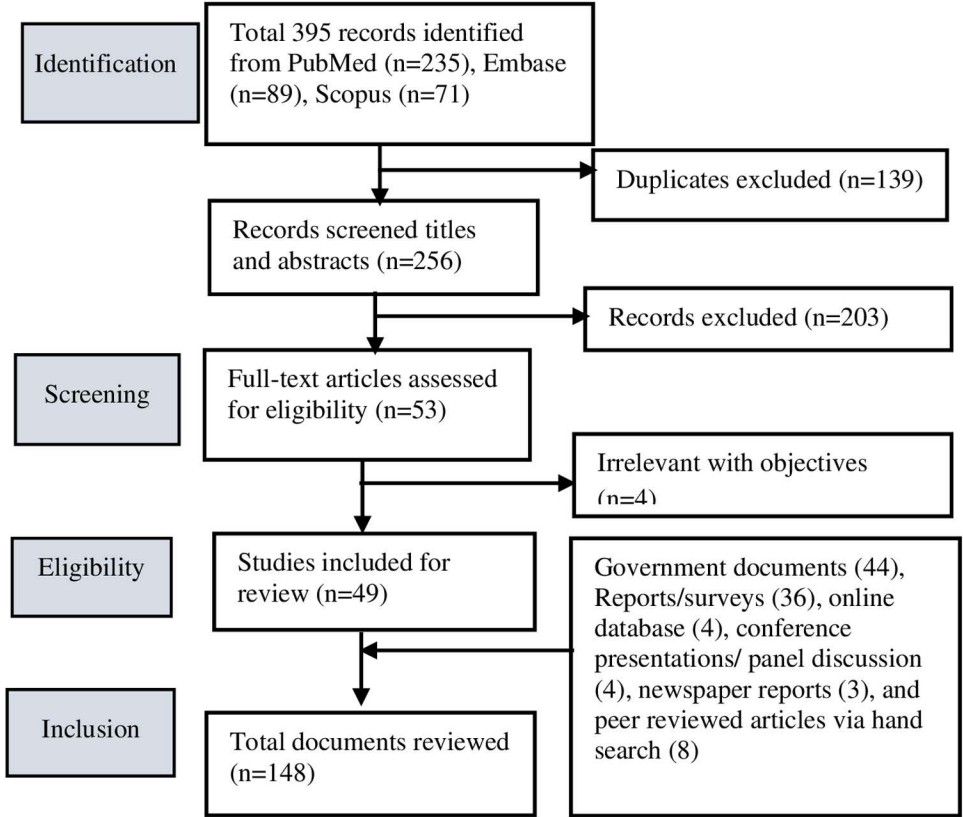

**Fig 1. PRISMA-ScR flow chart for selection of studies for the review.**

In addition to constitutional provisions, several key legislative frameworks have shaped health financing. These included the Health Insurance Act 2017, which introduced mandatory insurance and subsidized premiums for marginalized groups [23], the Public Health Service (PHS) Act 2018, which defined standards and criteria for free basic health services, emergency care, and public health governance[52]), Social Health Security Development Committee (SHSDC) (Formation) Order 2014, which mandated the establishment of SHSDC for designing and introducing the social health insurance (SHI) in Nepal [44] and Health Insurance Regulation 2019 (operationalized premium rates and claims processes) [56].

Since the 1990s, three National Health Policies (NHP) have been implemented: the NHP 1991, focused on donor-dependent health financing models [57]; NHP 2014 introduced SHI and decentralization, reducing donor reliance and aligning aid [61], while NHP 2019 emphasized domestic funding and the expansion of SHI [64]. The National health Insurance Policy 2013 also laid the foundation for universal coverage through a health insurance program [68].

The evolution of health sector implementation plans further reflects shifts in health financing priorities. The first NHS Program-Implementation Plan 2004–2009 was centered on donor funding, and centralized financing [94], while NHS Program-Implementation Plan II (2010–2014) piloted SHI, promoted decentralization, and introduced mixed financing models [69], The NHS Strategy (2015–2022) focused on federal resource allocation, SHI expansion, and digital integration [70], and NHS Strategic Plan (2023–2030) shifted health financing towards greater reliance on domestic resources, while strengthening and streamlining social protection schemes to promote sustainability and equitable access [25]. Finally, the National Health Financing (NHF) Strategy (2023–2033) aims to achieve tax-based financing for BHS and premium-based universal health insurance [26].

**Table 1. Overview of health financing in policy documents of Nepal: Status, issues and challenges.**

| Provision of policy documents | Status | Issues and challenges |
|---|---|---|
| **Constitutions, acts, regulations, and order** | | |
| The Constitution of the Kingdom of Nepal, 1990 (1990–2006)[19] • Focused on state obligations rather than individual rights | • Focused on basic health services in rural places by establishing health facilities [43]. | • No legal mechanism to hold the state accountable. • Health framed as a public welfare issue, not a fundamental right. |
| Interim Constitution of Nepal, 2007 (2007–2015) [21]: • Fundamental rights, health rights began shifting toward justice | • Focused on equity and social inclusion by ensuring free basic healthcare to marginalized communities [31]. | • Lacked strong enforcement mechanisms. • Interim status limited long-term policy implementation |
| Constitution of Nepal, 2015 (2015-) [22] • Health enshrined as a fundamental right (essential health services via BHS and other services through HI) | • Free basic healthcare guaranteed for all citizens in the constitution. • BHS program and health insurance implemented. • Health governance decentralized to provincial/local governments. | • Absence of financial and administrative federalism, despite federal structure with three tiers of government with legislative power. • Poor coordination among the three levels of government. • Insufficient funding to allocate resources and fulfil health rights. |
| SHSDC (Formation) Order, 2071(2014) [44] | • Basis for the design and implementation of NHIP | • Repealed by the HI Act [23] |
| Health Insurance Act 2017 [23] • Mandates membership in the health insurance program for all Nepali citizens. • Defines benefit packages, contribution amounts, co-payment, contracting mechanisms, and payment methods. • Establishes the Health Insurance Board as an autonomous body | • HI Board formed to replace SHSDC [23, 30]. • NHIP rolled out in all 753 local levels [30, 45] • Development of Health Insurance Strategic Roadmap (2024–2029) and guidelines for co-payment, referrals, and enrollment [46] • Introduction of insurance cards, subsidies, and co-payments starting in 2024 [30] • Decentralized structure with federal, provincial, and district offices • Budget allocations for subsidies and premiums. | • The Board lacks financial independence, with fragile leadership and issues like inadequate funding, unformalized structures, and a 20% co-payment affecting marginalized groups [14, 30, 47, 48]. • Only 46% of local levels have listed service providers, limiting access to health insurance [30]. • Limited participation of non-public facilities, with only 482 empanelled health facilities as of October 2024 [30, 49, 50] • Reliance on a fee-for-service model with varying rates [30] • Potential for moral hazards, fraud, and overuse of services [51] |
| Public Health Service Act 2018 [52]: • Define BHS as free and accessible promotional, preventive, diagnostic, curative, and rehabilitative health services divided into 10 components [52]. | • Standard treatment protocols for BHS and emergency services have been developed [53, 54]. | • Limited readiness of health facilities for providing BHS[45] • Less than 1% of public HFs provide all 41 BHS tracer components, and 75% lack basic equipment [55] • Quality standards at HFs not achieved [45] |
| Health Insurance Regulation 2019 [56]: • Define provisions in health insurance act for implementation. | • Provision of a positive/negative list, premium and benefit package, and health facility reimbursement in the HI program | • Although the regulation provides formal sector enrollment, this regulatory provision has not been implemented. |
| **Policies, plans** | | |
| NHP 1991 [57] • Foundation-building phase with a focus on vertical programs and primary care. • Mandated expansion of Basic Primary Health Services to the village level. | • Health Economics and Financing Unit established in 2002, and increased donor support integrated into sector wide approach (SWAp) for efficient resource allocation [43]. • Implemented community-based health insurance, SWAp, maternity incentives, free healthcare policy, and fee removal in Primary HFs (2007–2009), and community drug program [3, 43, 58–60] | • Challenges in retaining health workforce in rural areas and low investment in health [43] • Inefficiencies in enforcing health acts, policies, and action plans [43]. |
| NHP 2014 [61] • Transition toward equity, systems strengthening, and UHC (free basic health services, insurance policy, and provision of resource generation | • Provision of quality healthcare, financial protection, and increased accountability [14]. • Expanded HI for those 70+ and all families, including workers and the poorest [30]. | • Unclear roles of providers and purchasers[3]. • Fragmented funding caused inefficiencies and coordination issues [62]. • Limited power to negotiate prices, with no incentives for informal sector registration [3, 62, 63]. |

*(Continued)*

| Provision of policy documents | Status | Issues and challenges |
|---|---|---|
| NHP 2019 [64]<br>• Legally mandated free basic healthcare and emergency services<br>• Strengthened NHIP for broader coverage. | • Defined BHS services and public HFs for free BHS [52].<br>• BHS Operational Guidelines (2023) developed [55].<br>• Limited public HFs offering full BHS package [55].<br>• BHS Operation Guidelines rolled out [65]. | • Multiple SHS programs, duplication, and inefficiencies [3]<br>• Slow insurance enrollment and renewal [51, 66]<br>• Coordination gaps, underfunding, rural-urban disparities, and limited monitoring [55, 66, 67] |
| Health Insurance Policy 2013 [68] | • Limited reduction in OOP expenses.<br>• Health insurance act/regulation, HIB formation, NHIP in all districts [51]<br>• HIB's limited role as strategic purchaser.<br>• Growing fiscal imbalance in fund. | • Weak financial setup, flat contributions, no formal sector inclusion in NHIP.<br>• Lack of provider accountability, ineffective payment system, disjointed HI info system.<br>• Issues with poor household identification, formal sector enrollment, and scheme integration [14]. |
| **Plans and strategies** | | |
| NHS Program-Implementation Plan (2010–14) [69]<br>• Improve financial management.<br>• Actions: social and performance audits, transparency measures.<br>• Establish an audit oversight committee.<br>• Monitor health financing. | • High regulations, delayed budgets, duplication, and poor planning [70].<br>• Decline in health spending (6.2% to 5.1%) and low budget absorption (83%) [71].<br>• Social audits in HFs without budget authority and no financial committees initially [72].<br>• Delayed formation of financial committees (2012/13–2013) [73].<br>• Gradual improvements in work planning and budgeting [74]. | • Inequity in service use, health outcomes, and increasing disaster threats [70].<br>• Lack of integration in health protection schemes, weak capacity, budget constraints, and poor follow-ups [75].<br>• Top-down planning and budgeting, against decentralization, with central control over finances [74]. |
| NHS Plan (2015–2022) [70]<br>• Strengthen health financing and social protection.<br>• Develop health financing strategy, NHA, and resource allocation formula.<br>• Establish legal framework for health insurance and social protection. | • Health governance shifted from unitary to federal, with a constitutional obligation for free BHS [14].<br>• SHSDC formed in 2015 to implement HI [76].<br>• NHIP rolled out in 2016 to three districts [76].<br>• Delay in health financing strategy and NHA costing [77].<br>• BHS defined in public health regulation [14].<br>• Parallel health insurance schemes for formal sector employees [23, 78].<br>• HI Act 2017, HI Regulation 2019, and HI Board formed in 2018 [79].<br>• Social audit guidelines developed; audits in 1252 HFs [75, 80].<br>• Increased government health expenditure and per capita spending [81]. | • Inequitable resource distribution affects service use and outcomes [82].<br>• Fragmented social health protection schemes, poor coordination [29].<br>• Health insurance mandatory by law but voluntary in practice, low enrollment, and high dropout [14, 29].<br>• Health insurance coverage limited to 25% of the population [24].<br>• Overlapping interventions in social health protection schemes [83].<br>• Multiple interests and implementation challenges for HI laws, including HI Board autonomy [79].<br>• Inadequate and inefficient public health spending [29, 84]. |
| NHS Strategic Plan (2023–2030) [25]<br>• Increase domestic financing and efficiency.<br>• Improve development cooperation management and financing predictability.<br>• Provide free BHS in all settings, reform health insurance, and streamline protection schemes. | • Poor absorptive capacity of allocated budgets, poor implementation of performance-based resource allocation [29, 45].<br>• Slow domestic funding and fragmented donor support [29].<br>• Launch of e-initiatives, including a dashboard to monitor health equity [85]. | • Inadequate public spending and delays in primary hospital setup.<br>• BHS fails to meet healthcare needs, poor trust in free services, and fragmented health protection schemes.<br>• No consideration of demographic and epidemiological changes.<br>• Health budget cuts in FY 2023/24 may impact service quality[86]. |
| NHF Strategy (2023–2033) [26]<br>• Expand fiscal space and strengthen health financing governance.<br>• Ensure resource pooling and service purchasing via social security.<br>• Provide BHS based on rate-per-unit, geography, and disease burden. | • Pre-condition: financial governance, pooling, and purchasing mechanisms.<br>• Focus on BHS, with ongoing costing process.<br>• Strengthen emergency health services through proper resource allocation. | • Implementation underway.<br>• Pre-condition: financial governance, pooling, and purchasing mechanisms.<br> Focus on BHS, with ongoing costing process. |

Despite progress, such as reduced OOP costs, challenges persist, including low SHI enrollment, gaps in federal coordination, and inequitable access to services. The trends reflect a shift from donor reliance to domestic resource mobilization, from centralized to decentralized governance, and an incremental expansion of SHI, all aiming toward UHC. Policies have evolved from initial resource mobilization efforts to robust financial management and, ultimately, to the comprehensive integration of social health protections, culminating in formulating and implementing the NHIP.

### Health financing functions of social health protection schemes in Nepal

Table 2 presents different social health protection schemes/programs, and their health financing functions [see Table 2]. The health financing system in Nepal is predominantly tax-funded (through general taxation). Premium-based schemes like NHIP and private insurance are emerging as complementary mechanisms. Tax-funded programs (e.g., public financing, BHS) rely on central and local government pooling, and NHIP and the social security fund (SSF) demonstrate effective national-level pooling mechanisms. Tax-funded programs often lack a purchaser-provider split, limiting efficiency and accountability. They use input-based budgets and block grants, while NHIP and private insurance use performance-based payment mechanisms (e.g., capitation, case-based payments).

Table 3 outlines the health programs and evaluates their UHC dimensions. Nepal's health programs vary in service, population, and financial coverage. Public financing (through public providers) offers immunization, nutrition, and MCH services, with 76% outpatient utilization and free care for vulnerable groups. Free health care provides basic services but has been replaced by the the BHS program. BHS covers 10 primary care and preventive services and 98 medicines for all citizens. The Safe Motherhood Program covers maternal health services. The impoverished citizen treatment fund targets chronic diseases but has limited financial coverage. NHIP covers 28% of the population but faces high dropout rates. Private Insurance has low coverage, while the employee provident fund (EPF) and SSF support formal sector workers. Geriatric Health Services aid seniors but lack rural access.

### Service coverage

Key themes under this component include health system capacity for NCD-related health services, health system readiness to address emerging disease burdens, and constraints in service coverage.

**Health system capacity for NCD-related health services.** Nepal has implemented multiple health programs focused on maternal and child health and nutrition. Donor-supported disease control programs have reduced the burden of infectious diseases such as malaria, tuberculosis, and HIV/AIDS [14, 102, 103]. By 2019, COPD and IHD had replaced respiratory infections and CMNN-related deaths (1991) as leading causes of mortality [7]. Risk factors like air pollution, tobacco use, and poor diets have driven the rising NCD burden [7]. Sedentary lifestyles and unhealthy diets (e.g., high carbohydrate/fat intake) are now widespread. For instance, obesity among women of reproductive age surged from 0.2% (1996) to 4.1% (2016), while overweight prevalence rose from 2% to 20% in the same period [104].

Between 2000 and 2021, Nepal's UHC service coverage index increased from 20% to 54% [11] [Fig 2], with gains in maternal/child health (41%→77%), infectious diseases (7%→62%), and NCDs (20%→46%). However, health system capacity (e.g., human resources, hospital beds) improved only marginally (25%→37%) [11], leaving Nepal with the lowest health system capacity in South-East Asia [18, 105, 106].

**Health system readiness to address emerging disease burdens.** Injuries account for 7.6% of deaths, driven by RTAs (23/100,000) and suicide (9/100,000) [7, 18, 107]. Despite rolling out the package for essential NCD (PEN) interventions in 2016, preventive measures and multi-sectoral policies remain underprioritized [24, 108]. Diagnostic services for NCDs (e.g., diabetes, CVDs) have expanded since 2015, but health system readiness remains suboptimal [109]. Public health laws (e.g., air quality, food safety) and mental health interventions are inadequately enforced [4, 110-111].

**Table 2. Health financing functions in different social health protection schemes in Nepal.**

| Schemes | Enrolment | Actor/ implementor | Revenue collection and pooling of funds | Purchasing | Resource allocation and payment |
|---|---|---|---|---|---|
| **Public financing of public providers** [3] | Automatic | Federal, Provincial, and Local Governments | General taxation: Pooled in the federal and province-divisible fund. Municipalities receive conditional grants, fiscal equalization funds, and matching funds specified for health that cover non-salary inputs. Pooled donor assistance. | Fixed budgets for public facilities: No purchaser-provider split. Federal MoHP is responsible for national/ specialized hospitals. Provincial and local governments are both providers and purchasers of health services. | Input-based line-item budgets and block grants. Salaries are determined centrally and are paid to staff at different levels of government. Salaries are determined and paid for by locally hired staff. |
| **Free Health Care** | Automatic | Local Government: implemented in 2006 and later replaced by BHS [49] | General taxation: The central budget is pooled and distributed by the federal, provincial, and local governments (municipalities) | Local governments are both providers and purchasers of services up to 25-bedded public hospitals [49] | Capitation payments for outpatient services by local governments (NPR 5 per head). Block grants as received from the Government. |
| **BHS** | Automatic | Financed by the federal government and implemented by the local government[1] | General taxation: Pool fund Central budget reimbursement at the facility level | Local governments are both providers and purchasers | Capitation-based payments for outpatient services by local government . |
| **Safe Motherhood Program and free newborn care services** | Conditional cash transfer | Local Government and implemented since 2006 [49] | General taxation: Pool fund Central budget reimbursement at the facility level | MoHP purchases services from the public and enlisted private providers | Capitation-based payments from the MoHP. Cash incentives depending on the level of facility and cases of delivery. Year expense: 3 billion NPR[87]. |
| **Impoverished (deprived) Citizen Treatment Fund** | Conditional: hospitals and local government need to approve | Department of Health Services under Federal MoHP since 2015 | General taxation: Central budget managed by the MoHP | No purchaser-provider split. MoHP is both a provider and purchaser of the eight chronic diseases from enlisted public and enlisted private hospitals | Conditional grant Reimbursement up to NPR 100,000, Renal transplantation costs up to NPR 400,000 per patient [88]. Average annual expenditure : 8 billion NPR[87]. |
| **NHIP** | Mandatory by law but voluntary in practice | HIB at the federal level, and staff at the provincial level: initiated 2016 | Single pool at the national level managed by the HIB. Sources of funds are contribution amounts from insurers and block grants from the government. Premium collection per family NPR 3800 annualy [87]. | Purchaser-provider split. HIB purchases services from the public and enlisted private providers for enrolled members. Providers are empanelled public and enlisted private HFs | Fee for services and case-based payment. Expenditure per family 8350 (as of FY2024/2025) (only 24% of total 62.8 billion of NHIP expense) [87]. |
| **Voluntary Private Insurance** [3] | Voluntary | Private companies | Premium pooling: Separate pool at private health insurance providers | Individuals purchase services from hospitals out-of-pocket while insurance companies reimburse against the insurance policy. | Various practices at the individual insurance providers (reimbursement of the bills of enlisted services with deductibles/ copayments |
| **Enterprises Private Insurance** [3] | Voluntary | Private Enterprises | Enterprise contributions: Enterprises managed individually (No pooling) | Enterprises purchase health care services from providers on behalf of their employees or affiliates | Conditional reimbursement for enterprises |

*(Continued)*

| Schemes | Enrolment | Actor/ implementor | Revenue collection and pooling of funds | Purchasing | Resource allocation and payment |
|---------|-----------|--------------------|-----------------------------------------|------------|--------------------------------|
| Social Security Fund [3] | Mandatory by law | SSF | Contribution pooling: A single national pool for its members managed by the SSF. Yearly allocation: NPR 1.37 billion [87] | Purchaser-provider split. SSF purchases the services from both public and enlisted private providers on behalf of enrolled members | Payment to the facilities empanelled by SSF. Reimbursement is up to NPR 25,000 for outpatient, NPR 100,000 for inpatient, and NPR 1 million for chronic diseases. Covers members and their spouses (recently added). Covers all treatment expenses for work-related injuries and occupational diseases and NPR 700,000 for other accidents. 20% copayment for outpatient and inpatient services [89]. |
| Employee Provident Fund | Mandatory by law | EPF [90] | Contribution pooling: Single pool managed by EPF | Employees purchase services out-of-pocket from registered hospitals, and EPF reimburses the patient [91]. | Conditional reimbursement after reviewing claims. |
| Social Service Unit [92] | Automatic | Federal and Provincial Government | Conditional grant from general taxation: Pool managed by public hospitals through conditional grants received from the federal and provincial government | The Provincial Ministry of Health in each province is both purchaser and provider. | Reimbursement up to NPR 200,000 per patient [92]. Yearly expense NPR 2 billion[87]. |
| Geriatric Health Services [93] | Automatic | All three levels of governments | Conditional grant from general taxation: the federal and provincial government, as well as revenues generated by the hospital | Federal and provincial ministries are both purchaser and provider | Senior citizens aged above 70 years receive up to 50% discount on designated health services decided by the hospital management committee [93]. Yearly expense NPR 1 billion[87]. |

Note: Conversation rate: 1 USD = 138 (31 January 2025).

**Constraints of service coverage.** Deficiencies include poor infrastructure, workforce shortages, inequitable resource distribution, and delayed budgets [112]. High transport costs hinder obstetric care access [113], while low awareness of free services limits healthcare utilization among older adults [114, 115]. NHIP offers a limited service package, excluding certain types of care [116]. Service coverage and utilization remain low due to voluntary membership, limited public trust, restricted choices of HFs, drug shortages, long waiting times, low priority for insured patients, providers' negative attitudes, moral hazard, and an inadequate benefit package [28, 33-35, 116–121].

## Population coverage

Key themes under this component were access to HFs for services, utilization of health insurance, and shifting demographics and disease burdens.

**Access to HFs.** Nepal has a health system with an extensive network of HFs, including 234 public hospitals, 184 primary health care centers (PHCCs), 3,769 health posts, and 7,651 basic health service centers [122, 123]. Over 2,000 non-public facilities and approximately 28,000 registered pharmacies primarily provide curative services [122]. Despite

Table 3. Health programs and coverage as per UHC dimensions in Nepal.

| Programs | Service coverage | Population coverage | Financial coverage |
|---|---|---|---|
| **Public Financing of public providers** | Immunization, nutrition, MCH services, primary health outreach, and disease control programs are provided. User fees exist for most curative services, with exemptions for the poor, people with disabilities, and seniors aged 70 + . | In 2022/23, 76% of the population used outpatient services [49]; in 2021, 80% of children aged 12–23 months were fully vaccinated, and 43% of married women used modern contraception [95]; by 2023, 82% of people living with HIV were on antiretroviral therapy [96]. | Free health care services from public facilities. |
| **Free Health Care (replaced by BHS)** | Free outpatient services at the health posts, PHCCs and up to 25-bedded public hospitals. | Targets (all citizens- priority populations). | Free of care from all public facilities |
| **BHS** | The PHS Act 2018 and BHS Regulation 2020 define 10 primary care services [97], while the Department of Drug Administration manages 98 essential medicines provided free under the BHS package [45]. | All Nepalese citizens | Free of care from all public facilities for the defined interventions |
| **Safe Motherhood Program and free newborn care services** | Maternity care (normal, assisted, and surgical) and sick newborn care. | It targets all pregnant women and newborns, with 81% of women having 4 + antenatal visits and 79% delivering in institutions [95]. | Free from a program implemented in enlisted private facilities and all public facilities |
| **Impoverished (deprived) Citizen Treatment Fund** | The package covers eight chronic diseases (heart and kidney (renal failure) diseases, Alzheimer and Parkinson's, cancer, sickle cell anemia, head injury, and spinal injury) and includes 152 lab tests, 7 imaging services, 102 treatments, 26 heart procedures, 915 surgeries, and 1133 modern and 25 ayurvedic medicines[30]. | Targeted poor citizens, providing free treatment to 41,230 patients in 2023/24 (21,555 for cancer, 7,881 for kidney disease, 6,936 for heart disease, and 1,239 for spinal injury[49]. | Up to NPR 100,000 and NPR 400,000 for kidney transplant expenses[49]. |
| **NHIP** | Services (diagnostics, drugs, medical procedures,consultation, surgeries, and supplies) are available through 482 service providers [50]. | The scheme targets all 29 million Nepalese citizens, with over 8.6 million (28%) enrolled and the renewal rate is 54% (FY 2023/24) [45]. | The scheme offers NPR 100,000 for five-member families, with an additional NPR 100,000 for each elderly person over 70 years [14, 30] |
| **Voluntary Private Insurance** | Benefit package varies according to the private insurance policies | Targets all Nepalese citizens. Less than 1% of total population [98]. | Yearly renewal by insurers. |
| **Enterprises Private Insurance** | Depends on the enterprises insurance policy choice: lump-sum cash/ reimbursement to the medical bills | Targets workers of the enterprises. Achieved less than 1% [98]. | Selected private enterprises. |
| **Social Security Fund** | Out and inpatient department, diagnostics, surgical, medical and drugs. Separate maternity and accident-related scheme | Targets all formal and informal workers. The plan is to expand to spouses and children below 18 years of age. Over 1.9 million contributors registered on SSF[99]. Claim rate around 17% [100]. | The scheme covers NPR 25,000 for outpatient, NPR 100,000 for inpatient, and maternity expenses. It provides NPR 1 million for chronic diseases after two years of contribution and covers work-related accidents and up to NPR 700,000 for other accidents [30, 99]. |
| **Employee Provident Fund** | The scheme covers inpatient services (24 + hours), including treatment, diagnosis, drugs, and hospital bed expenses, and provides coverage for chronic diseases like cancer, heart attack, brain hemorrhage, kidney failure, and more[91]. | 582,000 formal workers registered in EPF (also includes their spouses since 2024) | The scheme offers up to NPR 100,000, with an additional NPR 1 million for 12 fatal diseases. NPR 166.6 million was spent on contributors' health in FY 2022/23, a decline from NPR 171.1 million in FY 2021/22 [78, 90] |

*(Continued)*

**Table 3.** (Continued)

| Programs | Service coverage | Population coverage | Financial coverage |
|---|---|---|---|
| **Social Service Unit** | The scheme provides general and specialized hospital services to the poor, gender-based violence survivors, people with disabilities, senior citizens, health volunteers, disaster victims, martyrs' families, and vulnerable indigenous groups; SSUs are operational in 94 hospitals nationwide [99] | Over 90,000 people are covered in 2023/24, with 51% female. Among SSU beneficiaries, 71% are poor, 18% are senior citizens, 5% are helpless, 2% have disabilities, and the rest are others [45]. | Support varies by case, with subsidies for medications and treatment up to NPR 2,500 for OPD, NPR 10,000 for inpatients at central/provincial hospitals, and NPR 1,200 for OPD and NPR 400 for inpatients at district hospitals. Over NPR 196 million was exempted through SSUs in 2023/24 [99] |
| **Geriatric Health Services** | Services available from 64 hospitals (>50 bedded)[45] | >49,000 senior citizens were provided with geriatric services in 2023/24[45] | Senior citizens receive the following subsidies: 50% for ages 60–69, 75% for 70–79, and 100% for 80–84 on specified services, with full coverage on all health services for those 85 and older[101] |

Note: Conversation rate: 1 USD = 138 (31 January 2025).

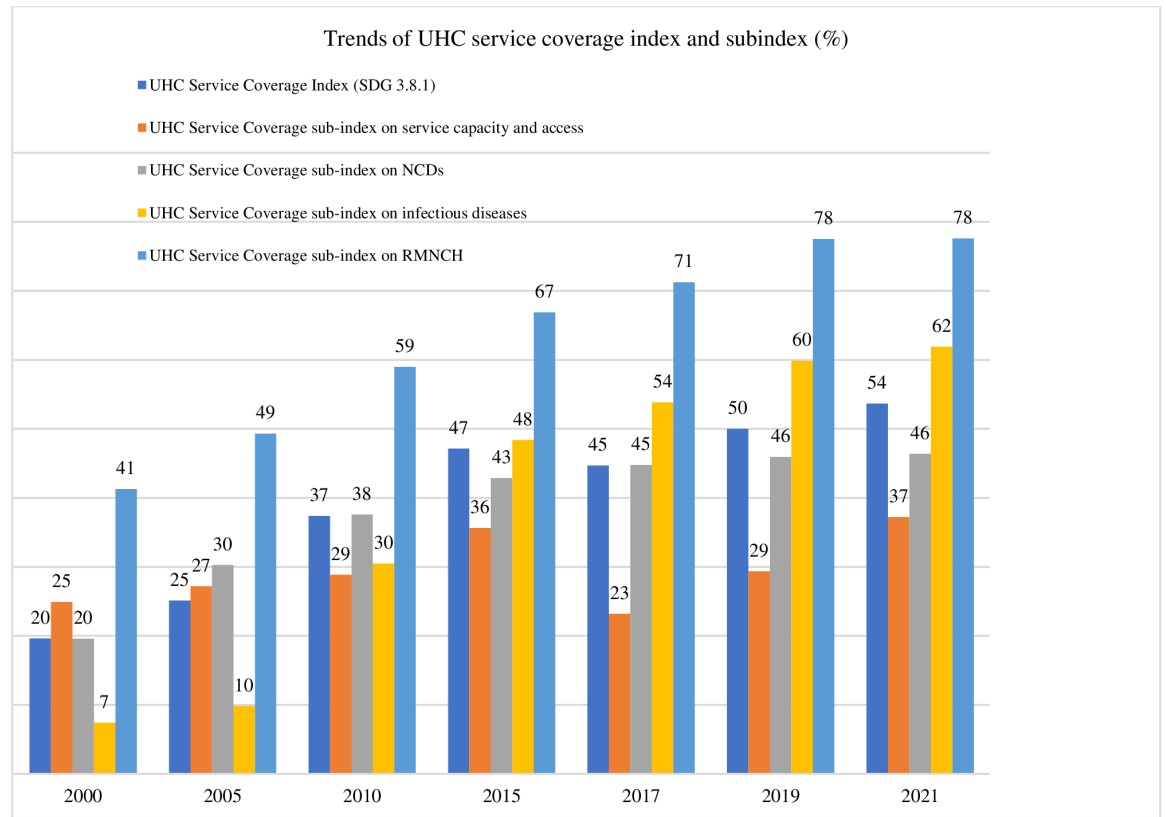

**Fig 2. UHC service coverage index for Nepal (Source [ 11]).**

plans to build hospitals in all 753 local units by 2020/21, only 322 were under construction by May 2024 due to funding gaps [45, 124]. Moreover, despite the government's efforts, persistent inequities continue to hinder progress toward UHC, exacerbated by limited physical access to HFs and low socioeconomic status (SES) [45]. Rural populations face stark inequities: 61% can reach a health post within 30 minutes [8].

**Health insurance utilization.** Coverage remains low but rose from 25% (FY 2022/23) to 28% (FY 2023/24) [125] [Supplementary Information, S1 Fig]. Renewal rates are suboptimal (54% over two years) [125]. Enrollment is higher among educated individuals (particularly household heads), advantaged castes/ethnicities, high-income households, and those with chronic illnesses or comorbidities [116, 126–128]. Health service utilization is greater among partnered individuals, female-headed households, families in accessible geographic areas, and people aged over 60 [103, 114, 115, 129]. Conversely, dropout rates are disproportionately high among affluent households, migrants, local political leaders, poor households, Dalits (a marginalized ethnic group), and households in rented accommodations or with perceived good health status [116, 120, 126-128, 130]. These disparities highlight systemic challenges in NHIP, including pro-rich bias and inequitable pricing structures [116, 131].

**Shifting demographics and disease burden.** Nepal's demographic structure is shifting toward an aging population, accompanied by a sharp rise in NCDs such as CVDs, cancers, and respiratory conditions (Fig 3) [8, 9], along with changing demographic structure [133]. This demographic transition has exacerbated the prevalence of NCDs and injuries, which now account for a growing share of the total disease burden, measured in DALYs [7, 24, 45].

Concurrently, Nepal's rapidly ageing population has shortened the window of demographic dividend while increasing demand for healthcare services, particularly for chronic and age-related conditions [Supplementary Information, S3 Fig] [132]. The upward trend in DALYs among older adults underscores the urgent need to expand NCD prevention and treatment programs, strengthen health literacy initiatives, and address systemic gaps in care delivery [7, 24, 45].

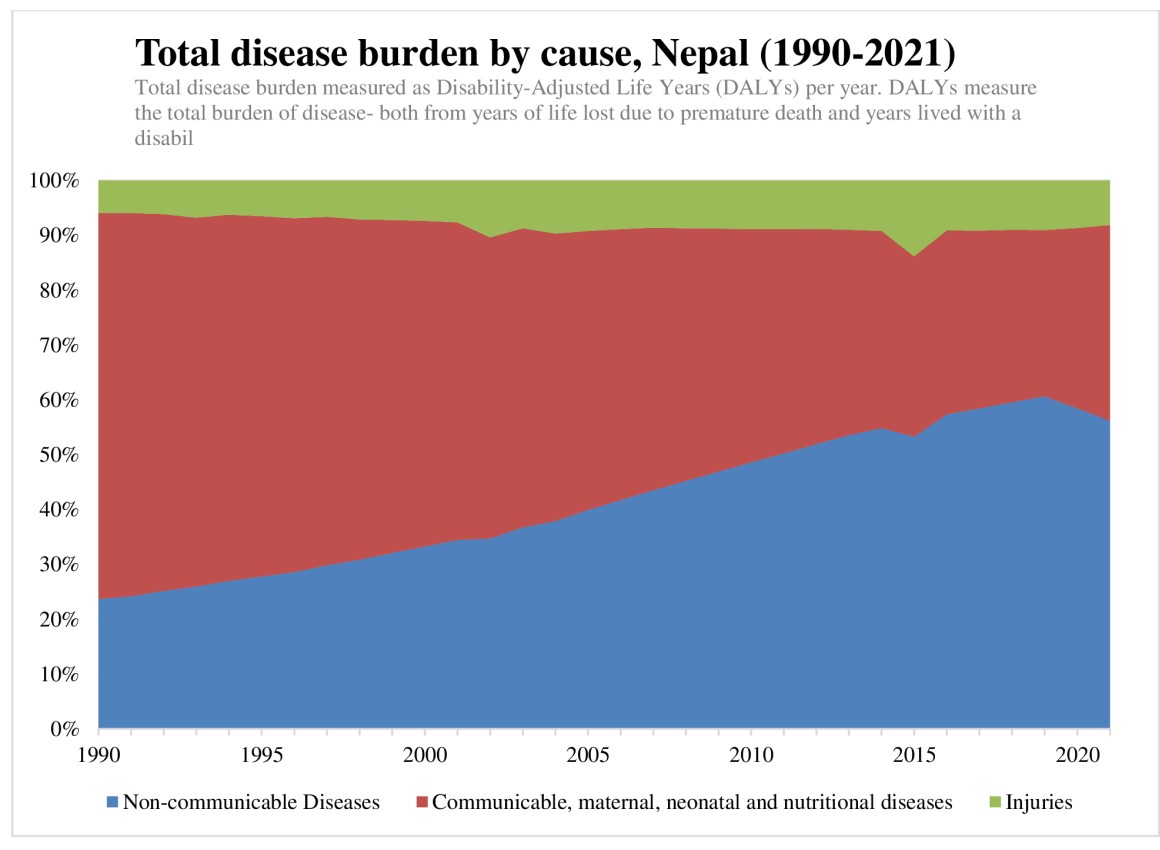

**Fig 3. Increasing NCD burden in Nepal over the last three decades.** Source: Our World in Data [132].

## Financial coverage

Key themes under this component included high OOP and financial burden due to NCDs, disproportionate budget allocation to local governments, and inequitable financial risk protection in healthcare.

**High OOP and financial burden due to NCDs.** The 2019/20 National Health Account (NHA) report highlights that OOP expenses in Nepal remain alarmingly high, accounting for 54.2% of CHE and reflecting systemic challenges such as inadequate government healthcare spending, ineffective social health protection schemes, and rising financial burdens on households [134]. Approximately 10% of the population faces catastrophic health expenditure (using the 10% threshold), while 1.7% are impoverished due to OOP costs (based on the $1.90 PPP threshold) [135]. These burdens disproportionately affect vulnerable groups, including the elderly, rural residents, low-income households, and individuals with chronic illnesses such as diabetes, asthma, and CVDs [136–141].

Expenditures on hypertension, CVDs, and cancers (e.g., lung, breast, cervical) constitute a significant share of CHE for both outpatient and inpatient services, with variations tied to age, socio-economic status (SES), illness severity, and perceived susceptibility [27, 142, 143]. Similarly, patients with cancers (lung, breast, cervical) face substantial financial strain, with medical costs accounting for 81% of total treatment expenses [139, 142]. Injuries (e.g., road traffic accidents) and chronic comorbidities drive catastrophic spending, with 70% of injury-related costs and 62% of NCD-related expenditures financed through OOP payments [77, 136, 138]. Industrial workers hospitalized for acute or chronic conditions face particularly severe financial strain: 13% experience CHE, and 42% resort to distress financing (e.g., loans, subsidies), primarily due to drug and diagnostic costs [137, 144]. Low-income households frequently depend on subsidies (43%) and loans (29%) to cover NCD treatments, exacerbating financial insecurity [8].

## Health financing inequities

As shown in Fig 4, OOP expenditures have remained above 50% of total health spending for over two decades, underscoring Nepal's regressive health financing system, which disproportionately impacts the poorest income quintiles [119, 145, 146]. Despite a total health expenditure of ~2% of GDP—far below the global average of 5%—the government struggles to increase budget allocations, improve absorption rates, or curb last-minute fiscal reallocations, hindering progress toward UHC [147–149]. Domestic health spending accounts for nearly one-third of CHE, while NCDs and injuries now represent 42.6% of CHE, up sharply in recent years [134, 150].

Persistent inequities in health budget allocations for NCDs and injuries further perpetuate high OOP spending (Fig 5) [151–153].

**Disproportionate budget allocation to local governments.** Over the last three fiscal years (FY 2022/23–2024/25), Nepal allocated NPR 103 billion, 84 billion, and 86 billion to local governments, respectively [45]. Fig 6 illustrates the distribution of health budgets across Nepal's federal, provincial, and local tiers of government during this period, revealing stark disparities in funding. Despite incremental increases in allocations to local governments, the federal level retains a disproportionately large share of the budget, underscoring persistent imbalances in fiscal decentralization [45, 125, 154, 155].

**Inequitable financial risk protection in health care.** Although Nepal's NHIP is legally mandatory, it functions as a voluntary program due to ineffective enforcement mechanisms, hindering population coverage expansion [14, 156]. Despite subsidies for underprivileged groups, Nepal's system for identifying marginalized and ultra-poor populations remains flawed [156]. Low insurance enrollment, high healthcare costs, limited risk pools and budget constraints further undermine efforts to support vulnerable groups [144, 157]. Key factors contributing to the NHIP's underperformance include: voluntary enrollment; conflicts of interest among health policymakers, politicians, private providers, and trade unions; ambiguous guidelines; contradictory legal provisions; the HIB's lack of a formal organizational structure; enrollment challenges; and its inability to competitively select providers or act as a strategic purchaser of services [27, 130, 156, 158, 159].

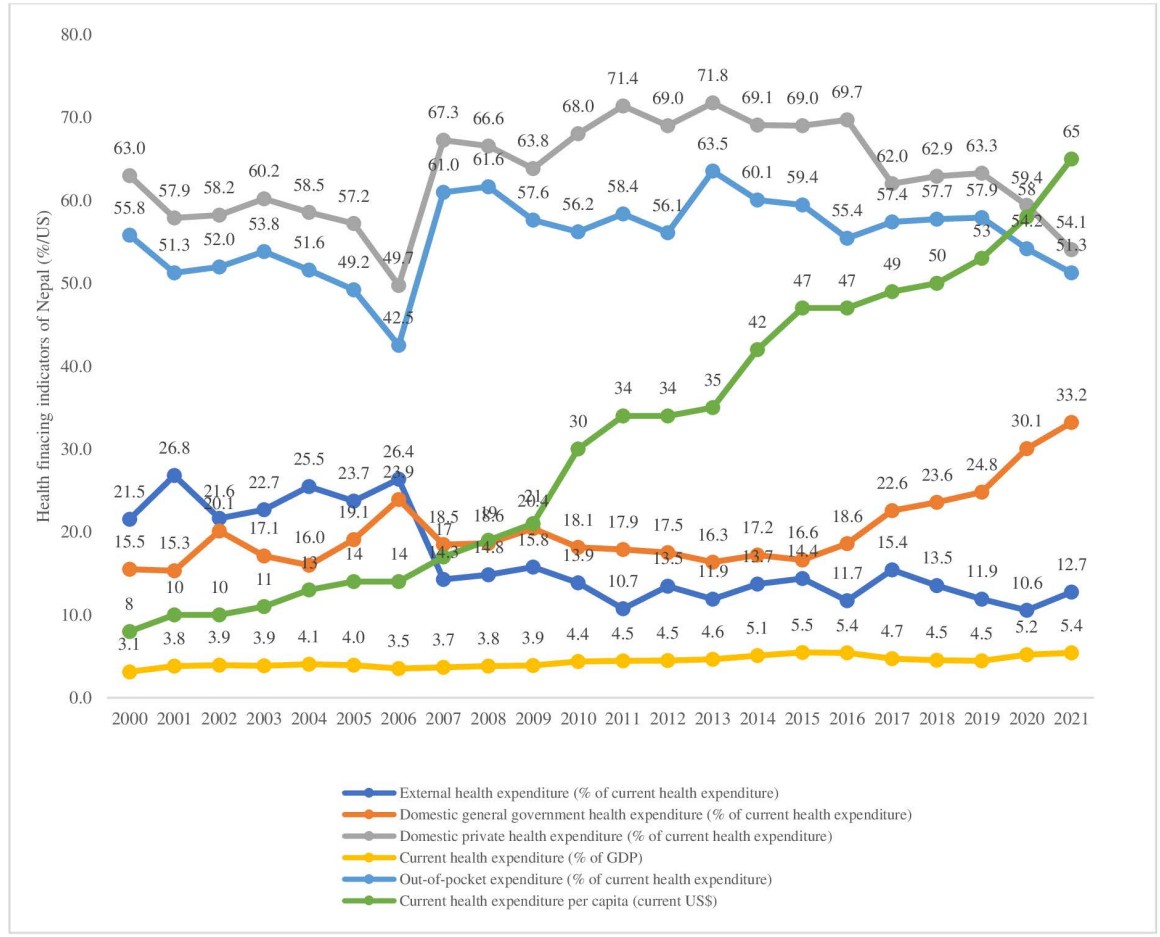

**Fig 4. Trend in health financing indicators in Nepal (2000–2021).** Data source: World Bank (https://data.worldbank.org/).

Rising healthcare costs in public and private sectors disproportionately limit financial access for marginalized groups [160]. Rural residents face crippling OOP expenses, with drugs constituting 85% of OOP costs among the poorest quintiles [10]. Non-medical expenditures (e.g., food, transportation) often exceed direct medical costs for individuals seeking care [156, 161, 162].

The PHS Act (2018) prohibits public funding for foreign medical treatments (Clause 30) [52]. However, the 2019 Standards Related to Treatment Expenses, Relief, and Financial Assistance introduced loopholes, restricting support to deprived individuals deemed to have "contributed to the nation" [163]. Politicians exploit these gaps to channel public funds—millions of rupees—toward overseas treatments for current/former elites, entrenching preferential treatment for advantaged groups [164] [See Supplementary Information, S4 Fig}. In other words, despite the prohibition of the PHS Act, political leaders continue to misuse public funds for personal health expenses via weaker legal tools like guidelines [165]. Although operational guidelines were developed to address this, they lack mechanisms to resolve systemic inequities or ensure accountability [166].

## Discussion

This study examines Nepal's health financing system through the lens of the UHC framework. The analysis highlights critical challenges affecting Nepal's progress toward UHC, including demographic and epidemiological shifts, as well as

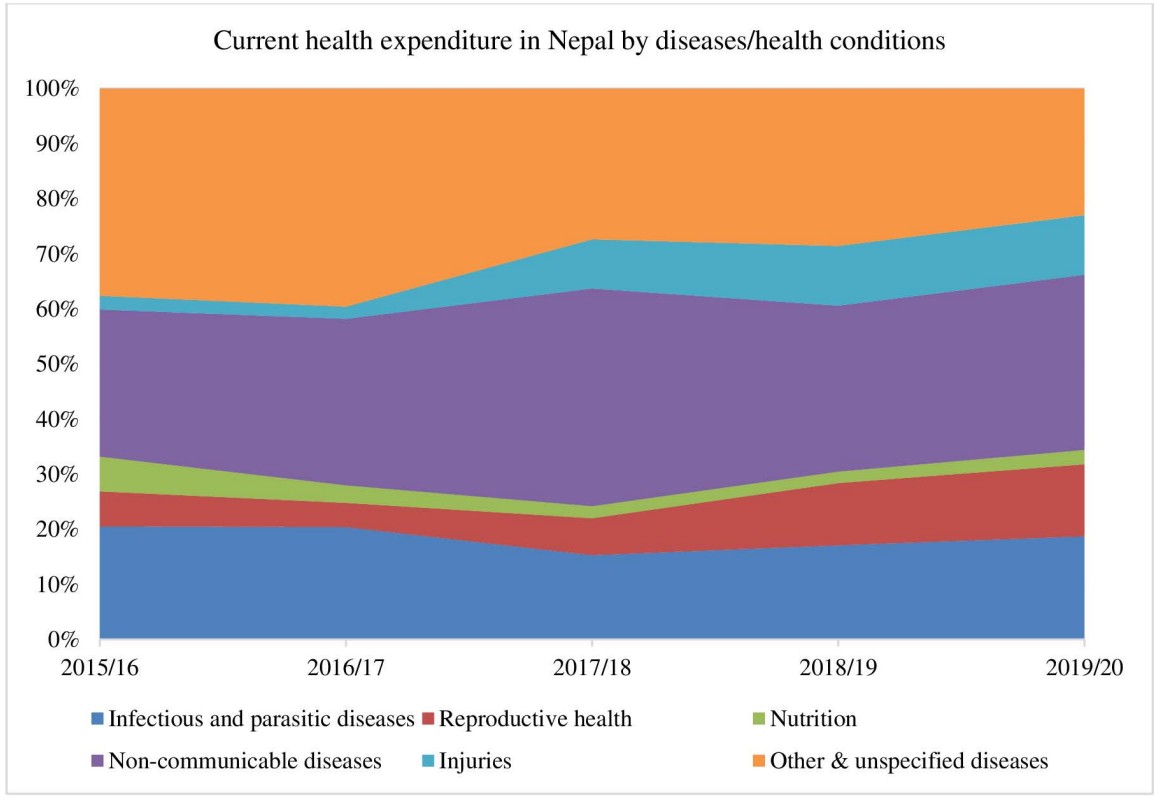

**Fig 5. Health expenditure by programs/diseases/health conditions in Nepal (NHAs 2015/16–2019/20) [** 151, 152**].**

gaps in policy responses. Nepal's health financing system has transitioned from a donor-dependent model to a focus on domestic funding, aligning with the constitutional shift that frames health as a fundamental right of citizens. The health financing system is not aligned with evolving demographics and disease burden (e.g., rising NCDs). Furthermore, fragmented social health protection programs, both within and outside the health system, result in insufficient risk pooling and limited financial protection for marginalized populations, particularly for high-cost NCD treatments. Existing health services programs have limited benefit packages and low population coverage for treatment and care beyond the BHS package. Limited population-level interventions for NCD prevention contrast with heavy spending on treatment subsidies for chronic diseases, reflecting misaligned priorities. High OOP healthcare expenditures coexist with low health insurance coverage, exacerbating financial hardship. Disparities persist between privileged and disadvantaged groups in accessing affordable care. Health financing policies are undermined by inadequate institutional arrangements (e.g., weak laws, unclear organizational structures, fragmented policies) across governance levels.

## Paradigm shift in health financing system and policy landscape in Nepal

Nepal's health system has undergone significant constitutional, legal and financial transformations. Initially framed as a state-led welfare initiative under the 1990 Constitution [19], health was later recognized as a fundamental right in the 2007 Interim Constitution [21], culminating with the 2015 Constitution's [22] emphasis on a federal, decentralized, inclusive health governance system aligned with global equity goals. Parallel shifts occurred in health financing, transitioning from donor-dependent, centralized models in NHP 1991 [57] to a priority on domestic funding and emergency health funds [64]. Current trends reflect a shift toward domestic resource mobilization, decentralization, and crisis preparedness [31, 124].

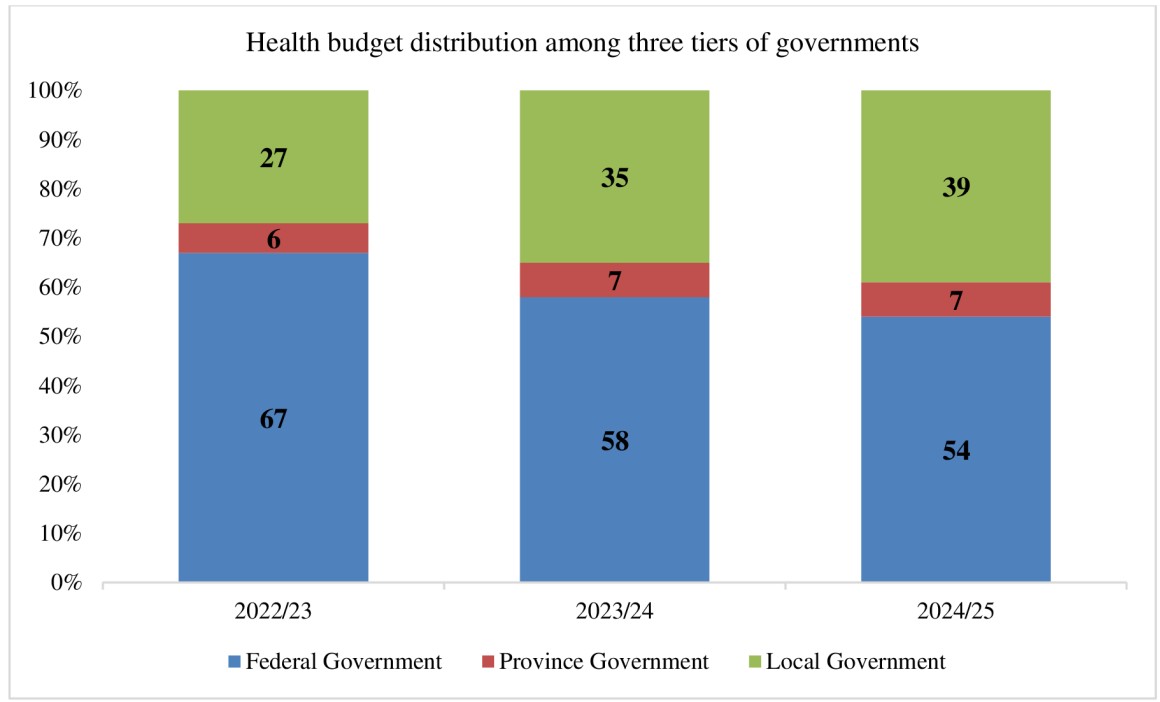

**Fig 6. Health budget distribution across Nepal's three tiers of government.** Source: Progress of Health and Population Sector 2023/24 (2080/81) [45].

However, implementation of these constitutional obligations poses significant challenges considering Nepal's economy and public sector budget constraints. For instance, Nepal's small economy, with a total GDP of $40 billion, is largely reliant on agriculture, services (tourism) and remittances [167]. Its informal sector, comprising unregistered small and middle enterprises, and subsistence farming, accounts for roughly 39% of economic activity as a share of GDP, largely in the agriculture sector [168]. This informality poses significant challenges, limiting tax revenue and formal job creation, and weakening the country's fiscal base. Additionally, Nepal faces a persistent trade deficit of up to 15.5% [169], while foreign grants and loans finance one-third of the national budget [169]. High recurrent costs consume two-thirds of the budget, leaving limited funds for development [170].

Tax-funded social security programs account for 16.6% of the total government expenditure and 4.9% of GDP [32], raising concerns about the sustainability of welfare schemes such as the old-age allowance [171]. Similar budgetary constraints impact the health sector, where long-term investments remain inadequate. Nepal's public health spending stands at less than 2% of GDP [14, 154, 172], far below the WHO-recommended 5% needed to limit OOP payments and achieve UHC [12]. This level of spendings also lag behind regional counterparts [13], highlighting a critical funding gap. More than three-quarters of government health expenditure is allocated to recurrent costs, highlighting chronic underfunding of capital investments [154]. This shortfall hinders infrastructure development, such as rural health centers and diagnostic facilities. Furthermore, Nepal's over-reliance on donor funding poses sustainability risks if aid diminishes [173, 174]. Ensuring long-term financial sustainability is further complicated by systemic inefficiencies, wastage, and corruption which undermine effective resource utilization [67, 175]. Addressing these challenges requires strategic reforms in health financing, improved fiscal discipline, and enhanced efficiency in resource allocation. Strengthening federal coordination and enhancing budget transparency are equally crucial to mitigating governance challenges, including erratic spending and corruption.

## Tax-based versus premium-based health financing models

Nepal is adopting both tax-funded and premium-based health financing models. The premium-based NHIP uses the family as the unit of enrollment and premium payment and is designed to reduce the OOP expenditure by providing services beyond tax-funded BHS [53, 55]. Evidence indicates that premium-based SHI models may increase OOP expenditures more than tax-funded programs in LMICs due to SHI's higher implementation costs and more limited coverage [176, 177]. In LMICs, contributory premium-based health insurance schemes have not consistently proven to increase revenues to the health sector or help countries achieve universal health coverage [176], while transitions to government financing increase life expectancy and reduce catastrophic health expenditure incidence [177]. Instead, tax-funded systems promote equity through cross-subsidization and prioritize rural primary care, whereas SHI risks exacerbating urban-rural disparities unless paired with substantial subsidies [178].

While expanding coverage to the informal sector remains a major challenge for most LMICs, especially those relying on contributory systems, countries such as Bangladesh, Indonesia, Thailand, Vietnam and Ethiopia have successfully initiated their SHI programs by first covering civil servants and formal sector workers, leveraging their institutional ties to tax systems [179–181]. Nepal's NHIP, however, was launched by first targeting households, which may have resulted in struggles to maintain sufficient coverage, achieve adequate risk pooling, and ensure long-term financial sustainability due to unpredictable revenue collection.

In Nepal's context, tax-funded health financing models could be more equitable than premium-based SHI due to widespread poverty and a large informal workforce [174]. However, Nepal's public health spending (<2% of GDP) is insufficient for sustainability without reforms like broadening the tax base or implementing earmarked health/sin taxes. Considering the size of the economy and public sector financing, reforming the current premium-based NHIP may not be feasible. Thus, a hybrid financing model that combines general taxation, payroll and member contributions, as well as health taxes and private or donor contributions, could be a viable option for Nepal. Thailand's National Health Insurance offers a blueprint: These schemes are predominantly non-contributory, financed by general taxation; with higher-income individuals contributing progressively through the income tax system, ensuring equity in financing. Formal sector workers are covered under the contributory Social Security Scheme (SSS), which is funded by mandatory payroll contributions (1.5% of salary each from employees, employers, and the government). Informal sector workers, who constitute a significant portion of Thailand's labor force, are covered under the Universal Coverage Scheme (UCS), with full cost government subsidies. Additional to this, charities and partnerships between public and private sectors are potential revenue sources for health [182, 183]. Nepal could draw on this model, as it has already introduced 2% payroll contributions (1% each from employees and employers). This should be designated as a health insurance tax at the base rate, supplemented by a progressive health insurance tax for high-income earners to pool funds for the sustainability of the NHIP. The government should also incorporate subsidized contributions for informal sector workers who represent 62% of Nepal's labor force [168, 184]. Engaging private sector providers through Public-Private Partnerships (PPPs) can help further optimize resource allocation and service delivery.

## Aligning health financing with disease burden

Nepal's health financing remains misaligned with its evolving disease burden, with disproportionate focus on CMNN issues despite the increasing burden of NCDs and mental health conditions. While, publicly funded initiatives, largely target CMNN interventions at both facility and community levels, structural weaknesses such as staffing shortages, fragmented governance, and inadequate fiscal space [24, 157] have hindered the integration of NCD prevention and treatment within the broader health system. Additionally, systemic underfunding and weak multisectoral collaboration continue to impede efforts to address NCDs [7, 108, 185] preventing a holistic approach to health financing. Expanding fiscal space for NCD interventions—such as screening programs (e.g., cervical/breast cancer) and integrated PHC services—could reduce premature mortality, DALYs, and OOP expenses [153, 186–188]. However, implementation gaps persist, exemplified by the poorly executed BHS program, which delays early disease detection [1, 189, 190].

Nepal's health system paradoxically prioritizes specialized hospitals over PHC, hindering NCD prevention [10]. This has created a situation where PHC facilities are being underutilized while tertiary hospitals are overwhelmed with patient flow. A "whole-of-society" approach is critical, one that includes integrating NCD services into PHC, expanding screening, and implementing community-based strategies (e.g., PEN interventions) to bridge gaps in early detection [153, 191, 192]. Other examples include ensuring at least one empaneled health facility per local government (410 of 753 lack such access) to provide health insurance services [32]. Redirecting resources to underserved populations is critical, addressing geographic and socioeconomic disparities through targeted subsidies and equitable resource allocation [108, 193]. Finally, multisectoral collaboration can potentially align policies with demographic shifts and leverage cost-effective interventions and global best practices [118, 136, 137]. Our analyses revealed that the UHC sub-index for service capacity and access and the index for NCDs remain weak, underscoring the need for targeted interventions. Expanding the scope of NCD care within the existing BHS package is essential to ensure comprehensive prevention, early diagnosis, and treatment. Strengthening the health workforce and adopting task-shifting strategies can enhance service capacity and improve accessibility, particularly in underserved areas.

### NHIP reforms for addressing financial risk protection in healthcare

Over 50% of health expenditures in Nepal are financed through OOP, driven by NCD medication costs, insufficient public funding, and unregulated private-sector fees [10, 45, 141]. Catastrophic health expenditure affects 10.3% of households, disproportionately impacting low-income and rural populations [141]. Existing social protection schemes, such as the NHIP and the impoverished citizen treatment fund, prioritize treatment over prevention and suffer from restrictive benefit packages (e.g., an NPR 100,000 treatment cap) and low enrollment (<30% coverage) [14, 31, 32, 49]. Stark inequities persist across wealth quintiles, ethnicities, and provinces, exacerbated by elite preferential treatment (e.g., politicians' medical funds) and fragmented risk-pooling mechanisms [52, 97, 193].

The NHIP faces structural issues such as coordination gaps between federal, provincial, and local governments, resource misallocation, infrastructure gaps, and workforce shortages that further hinder implementation [30, 32, 124]. Sustained reforms include improving intergovernmental coordination and enhancing funding efficiency—which remain critical to achieving UHC by 2030. To achieve this, the NHIP requires urgent reforms in expanding coverage caps, and enforcing mandatory enrollment to broaden risk pools [31, 32, 56, 47, 162, 194], replacing flat premiums with progressive tax contributions and subsidizing premiums for marginalized groups [30, 119, 130, 159]. A 20% co-payment is generally appropriate, but its determination should consider factors such as the premium amount, the total benefits received by the individual, a minimum premium for targeted groups, progressive premiums for accessing secondary and specialized services at government health institutions, and additional premiums for higher-level facilities [87]. Furthermore, digital transformation is vital to implement biometric enrollment, automated claims processing, and streamlined reimbursements to reduce fraud and improve transparency [30, 195, 196]. Governance autonomy is fundamental, and granting the HIB full operational independence from the MoHP strengthens strategic purchasing and accountability [30]. Service categorization (inpatient and outpatient caps), claims verification and evaluation to ensure clear claims, penalties for healthcare institutions or employees making false claims, a carry-forward system to prevent unnecessary treatments, integration of scattered programs to improve financial efficiency and eliminate redundancy, and promoting financial transparency and good governance could collectively enhance the NHIP's effectiveness and accountability [87].

### Maximizing risk pooling by minimizing fragmentation and duplication

Nepal's social health protection system is characterized by fragmentation and duplication, primarily between the Health Insurance Program, SSF, and the free health care program alongside several other social protection schemes. These fragmentation presents challenges, including high administrative costs, inefficiencies in financial resource management, and complications in ensuring comprehensive coverage, strategic purchasing and redistribution of resources. Programs

like the Free Health Care Program (tax-funded) and the Aama Program, which fund services already covered by NHIP, demonstrate duplication and overlap. Currently, multiple parallel schemes incur significant financial outlays: the Poor Citizens' Treatment Fund spends NPR 6 billion annually on chronic diseases, the Aama Program costs NPR 3 billion, Social Service Units use NPR 2 billion, free treatment for children and seniors accounts for NPR 1 billion, dialysis and chronic disease treatments cost NPR 2 billion, and the SSF maternity scheme claims NPR 1.37 billion [87]. Clearly, such fragmented schemes are not effectively serving Nepal's needs and require urgent reform. Merging these fragmented schemes into a single risk pool, or harmonizing them to ensure complementarity could enhance the redistributive capacity and strategic purchasing power for gains in efficiency and equity, but the success hinges on strong political will and legal reforms [11]. Global examples demonstrate the feasibility of such reforms. Thailand's UCS established in 2002, merged the medical welfare scheme for low-income groups and contributory voluntary health card schemes for households, providing safety nets for populations attending primary-care units. Meanwhile, the tax-finance medical benefit scheme for public employees and contributory social security scheme for private employees operate in complementarity with UCS [182]. Similarly, in 2003, Korea consolidated its various risk pools into a single national health insurance system, streamlining both management and operational structures[197]. Additionally, several other LMICs, such as Ghana, Indonesia and Vietnam have also made efforts toward integrating or harmonizing their multiple programs to improve financial protection for their population[180]. Nepal can draw lessons from these experiences to develop a more integrated, sustainable, and equitable health financing model with broader risk pools.

**Raising revenues for health.** Nepal's public financing on health remains low, covering only 33% of the CHE, which places increased pressure on households due to high OOP spending [13]. Ensuring adequate fiscal space for health financing requires the country to diversify and expand its revenue sources. Domestic resource mobilization- through measures such as sin taxes on health-harming products and broadening the tax base- offers a viable approach to increasing health sector funding and combating NCDs [174, 198]. For instance, out of a total of 6,666,937 households (20% of which belong to poor families whose premiums can be covered by the Government of Nepal), with a contribution of NPR 10,000 per household and a progressive tax-based premium, the total premium collected amounts to NPR 66 billion [87]. The funding sources include NPR 16 billion from program integration, NPR 5 billion from local and provincial governments, NPR 5 billion from a sin tax, and 25% of the total expenditure covered by the HIB [87]. Many countries worldwide have implemented special taxes, funds, and budgets for health, with many directly allocating a portion of excise taxes on tobacco, alcohol, and other health-damaging items, as well as taxes from salaries and even lottery tickets, to fund health expenditures [87]. Countries like the Philippines have successfully leveraged sin taxes on tobacco, alcohol, sugar-sweetened beverages and other ultra-processed foods to generate billions of dollars since 2012 [199]. A significant portion of this revenue has been allocated to the health sector, contributing to a fivefold increase in subsidies for PhilHealth, the national health insurance scheme, thereby strengthening UHC [198, 200]. Similarly, Thailand, South Africa, and Vietnam have also improved sustainable health financing through dedicated taxation on harmful products [182, 201].

**Resource allocation.** Funds should be directed towards high-impact areas like primary care and preventive services, focusing on vulnerable populations [29]. A shift from hospital-centric spending to decentralized primary healthcare is essential, particularly in rural areas, as a significant portion of care costs is also borne indirectly through expenses such as food, transportation, and loss of workdays while seeking care [10]. As seen in Brazil's decentralized care model, data-driven tools, governance reforms focusing on PHC, and balancing equity with sustainability are key to improving access and capacity [202, 203]. Improving healthcare delivery requires enhancing efficiency, utilizing technology, prioritizing community-based approaches, and evaluating the long-term impact of policies and investments, while strengthening monitoring, accountability, governance, and transparency in healthcare spending [204]. Additionally, implementing performance-based financing, focusing on health promotion and preventive care, developing prevention-focused policies, and raising public awareness of health and well-being are crucial for sustainable progress and overall public health improvement. To support these efforts, allocating a larger proportion of the national budget to local governments

is essential, as they play a central role in managing primary healthcare and implementing community-based health interventions.

**Purchasing of health services.** Purchasing health services needs to be based on performance, quality, and cost-effectiveness rather than input-based budgeting. This requires shifting from input-based funding models, such as fixed salaries for health workers regardless of performance, to output- or outcome-based contracts, and bulk purchasing of commodities. Strategic purchasing models prioritize quality and cost-effectiveness, like Thailand's performance-based contracting [197] or Brazil's selective contracting [203]. The enrollment process and co-payment methods can help reduce treatment costs, while any unspent amount can be carried over to the next year, alleviating unnecessary treatment pressure; reinsurance can be arranged for serious patients, and resources could be effectively utilized by controlling expenditures and developing a system to scrutinize services and claims for enrollment or non-enrollment [87].

### Implications of study for policy, program, and research

The study highlights key implications for policy and research, emphasizing the need for evidence-based decision-making, program improvements, and targeted interventions. It underscores the importance of addressing gaps in implementation, enhancing resource allocation, and fostering collaboration to achieve sustainable outcomes in policy and program effectiveness. Fig 7 presents the synthesized summary of implications of this study in policy, program, and research [Fig 7].

### Strengths and limitations of the study

This review of Nepal's health financing systems landscape utilized secondary data from multiple sources and was analyzed through a UHC lens as its guiding framework. The findings were reviewed and validated by health financing professionals in Nepal, enhancing the relevance and comprehensiveness of the overview. Despite these strengths, the study has limitations. Although expert validation was incorporated, the perspectives gathered may not fully represent the entire spectrum of stakeholders in Nepal's health financing system. Consequently, the findings could be subject to bias based on the viewpoints of those consulted. Additionally, the quality and depth of secondary data sources varied, potentially influencing the interpretation of results. Nevertheless, this review offers valuable insights to inform future research and policy discussions on health financing in Nepal.

### Conclusions

Nepal's health financing policies are progressive and aligned with the aspiration of UHC, yet critical gaps persist in multiple dimensions. A disproportionate resource allocation to address the growing burden of NCDs and injuries, fragmented social health protection schemes, underfunding (as a percentage of GDP), and limited fiscal space hinders equitable access to quality care. The current health financing system remains inefficient and financially unsustainable, struggling to adapt to Nepal's demographic and epidemiological shifts while maintaining adequate risk pools for viable schemes. To enable Nepal's progress toward UHC, we propose the following policy reforms: 1) Enact legal reforms to merge existing social health protection schemes into a unified or fewer schemes, ensuring complementarity with the SSF to enhance efficiency in pooling and purchasing functions.2) Enhance domestic health financing – increase health funding (≥5% of GDP) by implementing a mix of revenue-raising strategies, including the universal enforcement of the existing 2% payroll contribution from employees and employers, adopting progressive health taxation for high-income earners, and earmarking sin taxes for health. 3) Reform the NHIP- make health insurance mandatory for formal sector employees while subsidizing coverage for informal sector workers; and strengthen NHIP operations through strategic purchasing, expanding the geographical coverage of empaneled health facilities, digitizing claims, broadening benefit packages, and integrating private

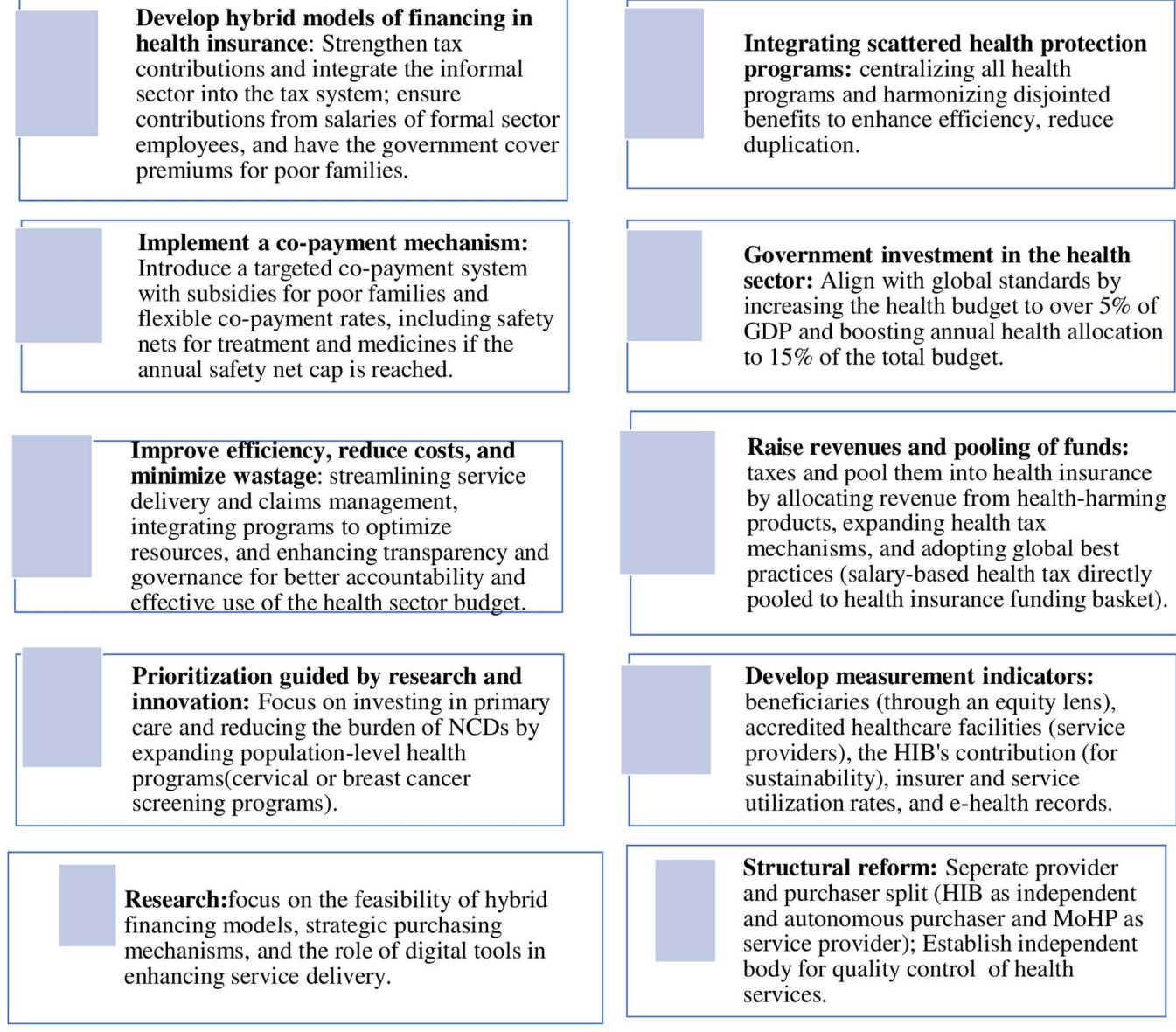

**Fig 7. Summary of implications of study findings for policy and program.**

providers with regulated pricing to reduce dropouts and enhance service access. 4) Improve equitable public financing and fiscal discipline– ensure equitable and needs-based allocation of health financing across federal, provincial, and local governments; and strengthen budget allocation and execution mechanisms to effectively respond to emerging epidemiological trends, including the increasing burden of NCDs and injuries, while maintaining adequate investment in CMNN care. Further research is needed to assess the feasibility of hybrid tax and premium based insurance models and their financial viability in Nepal. Additionally, studies exploring the optimization of strategic purchasing mechanisms, efficiency in health expenditure, and the role of digital health innovations in enhancing service delivery and financial protection are warranted.

## Supporting information

**S1 Table: Preferred Reporting Items for Systematic reviews and Meta-Analyses extension for Scoping Reviews (PRISMA-ScR) Checklist.**
(DOC)

**S2 Table: Search strategy.**
(DOC)

**S3 Table: Selected documents and reports in the review.**
(DOCX)

**S1 Fig: Health financing policy trajectory of Nepal.**
(TIF)

**S2 Fig: Health insurance coverage in Nepal.**
(TIF)

**S3 Fig: Disease burden by age, Nepal (1990–2021).**
(TIF)

**S4 Fig: Healthcare expenditures for political leaders and power elites during 2013–2022 in Nepal.**
(TIF)

## Acknowledgments

The authors express gratitude to the presenters and panelists who shared their views and opinions during public forums and conferences organized by the Nepal Health Research Council (NHRC), including: The Eighth National Summit (April 11–12, 2022), specifically Plenary Session I: Strengthening Social Health Protection in Nepal: The Role of Evidence toward Universal Health Coverage (Speakers: Suresh Tiwari, Hema Bhatt, Shambhu Gyawali, and Swati Srivastava); The 9th National Summit 2023, Parallel Session 8: Evidence on Social Health Protection for Strong Institutions (speakers: Suresh Tiwari, Damodar Basaula, Shambhu Gyawali, Geha Nath Khanal, Bikesh Bajracharya, Mukesh Adhikari, Shiva Raj Adhikari, Madan Upadhyay, Laxmi Ghimire, Devi Prasad Prasai, and Dinesh Kafle); The 10th National Summit 2024, Plenary Session 1: Health Policy Reform and Financing (speakers: Shiva Raj Adhikari, Krishna Paudel, Guna Nidhi Sharma, Achyut Raj Pandey, Ravikant Mishra, and Pratik Khanal); A seminar organized by the National Planning Commission, Nepal (April 8, 2022), featuring Krishna Paudel and Shiva Raj Adhikari. This paper expands on a concept presented by the first author during a webinar organized by the Nepal Public Health Association (April 16, 2023). The authors also extend thanks to PhD candidates in health economics and financing—Umesh Prasad Bhusal (University of New South Wales, Australia), Geha Nath Khanal (Canterbury Christ Church University, UK), and Mukesh Adhikari (University of North Carolina, USA)—for reviewing the final manuscript.

## Author contributions

**Conceptualization:** Resham B Khatri, Mihajlo Jakovljevic.

**Data curation:** Resham B Khatri, Pratik Khanal, Dipendra Singh Thakuri, Prabesh Ghimire.

**Formal analysis:** Resham B Khatri, Pratik Khanal, Dipendra Singh Thakuri, Prabesh Ghimire.

**Investigation:** Resham B Khatri, Dipendra Singh Thakuri, Prabesh Ghimire.

**Methodology:** Resham B Khatri, Pratik Khanal, Mihajlo Jakovljevic.

**Project administration:** Resham B Khatri.

**Resources:** Resham B Khatri, Pratik Khanal.

**Software:** Resham B Khatri.

**Supervision:** Mihajlo Jakovljevic.

**Validation:** Resham B Khatri, Pratik Khanal, Dipendra Singh Thakuri, Prabesh Ghimire, Mihajlo Jakovljevic.

**Visualization:** Resham B Khatri, Pratik Khanal, Dipendra Singh Thakuri, Prabesh Ghimire.

**Writing – original draft:** Resham B Khatri, Dipendra Singh Thakuri.

**Writing – review & editing:** Resham B Khatri, Pratik Khanal, Dipendra Singh Thakuri, Prabesh Ghimire, Mihajlo Jakovljevic.

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
