## [Decision Letter · Decision Letter 0]

20 Dec 2024

Dear Dr. Khatri,

Thank you for submitting your manuscript to PLOS ONE. After careful consideration, we feel that it has merit but does not fully meet PLOS ONE’s publication criteria as it currently stands. Therefore, we invite you to submit a revised version of the manuscript that addresses the points raised during the review process.

We look forward to receiving your revised manuscript.

Kind regards,

Himanshu Sekhar Rout, PhD

Academic Editor

PLOS ONE

Journal Requirements:

2. Please include captions for your Supporting Information files at the end of your manuscript, and update any in-text citations to match accordingly. Please see our Supporting Information guidelines for more information: http://journals.plos.org/plosone/s/supporting-information .

Reviewers' comments:

Reviewer's Responses to Questions

**Comments to the Author**

1. Is the manuscript technically sound, and do the data support the conclusions?

Reviewer #1: Yes

Reviewer #2: Partly

Reviewer #3: Yes

2. Has the statistical analysis been performed appropriately and rigorously?

Reviewer #1: N/A

Reviewer #2: N/A

Reviewer #3: Yes

3. Have the authors made all data underlying the findings in their manuscript fully available?

Reviewer #1: No

Reviewer #2: No

Reviewer #3: Yes

4. Is the manuscript presented in an intelligible fashion and written in standard English?

Reviewer #1: Yes

Reviewer #2: Yes

Reviewer #3: Yes

Reviewer #1: Thank you for the manuscript. The study is relevant and contains evidence that is essential for countries seeking to progress towards UHC. Nepal is an excellent case of what is happening in several LMICs. Nepal has made great progress in fighting infectious diseases but seen a surge in non infectious health conditions and diseases. This study provides insights about what transpired on the health financing policy front in the country given the demographic and epidemiological shift the country is experiencing.

Below are my comments:

Reviewer report

Title: Financing of health policies and programs amid demographic and epidemiological

transitions in Nepal: a scoping review and synthesis of evidence using universal health

coverage framework

Main text

Introduction

1. The health system of Nepal was well described in the introduction. However, the introduction lacks a description of how the epidemiology has changed in the country. Could you provide details of the changes in the disease burden the country has experienced? The 2019 burden of disease report may be helpful. https://nhrc.gov.np/wp-content/uploads/2022/02/BoD-Report-Book-includ-Cover-mail-6_compressed.pdf

2. Page 5, Lines 100-101 “Nepal is afflicted by a double burden of both high infectious and non-communicable diseases (NCDs).” This statement is a contradiction of what you stated in the abstract (transition to NCDs etc but not dual burden). Clarify, for consistency.

3. The last two paragraphs in the introduction seem disjointed yet they are related. Including information on how the 2015 constitution of Nepal reflected the UHC ambitions and whether there are aspects of UHC that are not addressed by the constitution is likely to be beneficial to the reader.

4. The research gap that the study sought to address is not well elaborated in the introduction. You state that “Nonetheless, there is limited discussion about how health financing policies and strategies are in place to address these issues.” How limited is this discussion? Has there been any literature about the same? What financing policies are these, anyway? Have some studies reviewed some of these policies ready? And if yes, what did they find or what was the scope of the review? How different are they from this study? You should consider addressing these questions to give the reader a clear and well elaborated picture if the existing research gap that you are trying to fill with this study.

5. In the last two sentences of the introduction, you mention what you have done in this study and the likely implication. Rephrase the sentences and clearly state the aim of your study reflecting the scope (the status and challenges of financing of health policies and programs towards universal health coverage).

6. Ensure that the aim in the introduction and abstract are well aligned/the same.

Methods

7. The methods section is overly summarized which may have resulted into omission of relevant information for example in the search strategy: how were the search terms identified, were the search strings customised to the databases, was the search automated? What platforms were used if any? Concerning the selection of the articles: who did the selection, was it done by one or more people and how? What was the process for selecting the articles, was any platform used? Etc. Data extraction: how was the data extracted? Was it automated or done by individuals, did they use a template or program? For reproducibility, a more detailed methods section should be provided.

8. The study states that PRISMA-ScR guidelines were followed. It is great that the checklist was attached. The structure of the methods section should be aligned with the requirements of PRISMA-ScR guideline. This will also enable you have a more detailed methods section making it easy to follow. This could be done by including subheadings reflecting the main components of the guideline.

9. Specify the reference for the WHO UHC framework used as there are a number of variations for example a recent one http://www.uhcwpr.info/. Please include a website as part of the reference. Page 6, lines 132-136

10. Replace “and” with “or” in the statement “… in Nepali and English …” Page 6, line 137

11. Clearly state the key words used in the search. Were they “Health financing”, Universal Health Coverage” and “Nepal”?

12. State the date when the search was date.

13. Include details of the strategy used to search for grey literature both online and offline (if done).

14. Attach as supplementary material, the search strings used in searching the databases or platforms used.

Results

15. Include a table of characteristics of the included articles, and records

16. You applied the WHO UHC framework (UHC cube), I would suggest that you reflect this in table 1 too by splitting both the status and Issues and challenges columns into three columns each to reflect the three dimensions of the UHC framework.

17. Table 1: Some of the statements are general and could benefit from making them more specific. For example, when you state inequity, what aspect are you talking about, who are the disadvantaged, is it the poor, is it women? etc. When you mention poor public satisfaction with CBHI, what aspect of CBHI were people unhappy with, was it the service package, or something else? It will be helpful to be more specific in the Issues and challenges column.

18. Table 1: I would have expected you to include the National Health Insurance Act 2017 as one of the major health financing policies on the list in table 1. Was there a reason of leaving it out?

19. Table 2: Include a specific column for the name of the program that you are referring to in each row, separate from the actor(s) managing the program. This will make the table clearer and easier to follow.

20. Table 2: The scope of your study did not include functions but programs and policies. Please clarify why you are including health financing functions in the table.

21. Some of the sections provide explanation of the results. The explanation should be moved to the discussion. Page 20, Lines 160 to 171. Please cross check pages 15 to 20 to exclude the possible explanations provided and move them to the discussion section.

22. The results section seems to have references that were not part of the articles included in the study for example reference number 97 (IQAir. Air quality in Nepal Air quality in Nepal 2022). Such references and the respective statements they support should be moved and considered for the discussion section.

Discussion

23. The discussion explains the results to a great extent but misses comparing the findings in this study with similar studies that were done in Nepal or comparable settings. It will be great to include this comparison in the discussion.

24. One of the intentions of your study is to see whether there has been a mismatch between the health financing policies and the shifting demographics and disease burden in Nepal. However, this has not been discussed much and neither commented about in the first paragraph of discussion. Consider including a sub section on whether the policies are responding to the demographic and epidemiological shift and add a comment about the same in the first paragraph of the discussion section.

25. The sub section entitled “Policy and plans versus translation into programs and services” has content that does not fully reflect the title. The current content has elements beyond the scope of the title (page 22, lines 207-221) and omits other relevant content, it would be great to provide more relevant examples. Some of the content especially regarding the details on financial protection my fit better in the subsections that follow.

26. Some of the subsections should be considered for merging as they discuss a similar area for example “Health insurance program in limbo”, and “High OOP and catastrophic expenditure”.

Strength and limitations

27. Not interviewing people is part of the design of a scoping review in most cases so there is no need to mention it.

Reviewer #2: Title:

The current title is informative but overly lengthy, which may impact readability. We suggest revising it to "Health Financing Amid Nepal's Demographic Shifts: A Scoping Review Using the Universal Health Coverage Framework." This revision improves brevity, maintains clarity, and enhances reader engagement while retaining all critical elements and better aligning with the paper's scope.

Abstract:

The abstract is overly lengthy, with excessive background detail overshadowing the methodology, findings, and recommendations. We suggest condensing the background to focus on critical transitions (e.g., rising NCD burden, financial challenges) and aligning it more closely with the study's objectives. Prioritizing key results with quantifiable data (e.g., NHIP enrollment rates, budget allocation gaps) will improve clarity and impact. Recommendations should include actionable insights to demonstrate the study's practical contributions.

Keywords:

The keywords are relevant but can be optimized for precision and searchability. Terms like "policy and strategy" are too broad and should be replaced with "health financing policies" to better capture the paper's content. Combining "service coverage" and "population coverage" into "universal health service and population coverage" would streamline the terminology. Adding "Nepal" as a keyword would improve geographic specificity and relevance.

Introduction:

The introduction provides essential context but is overly detailed, with outdated references that reduce its relevance to current challenges. We recommend replacing older references with recent studies and reports (from the past 5 years) on NHIP, UHC progress, and health financing trends. The section would benefit from a focused problem statement and earlier presentation of study objectives, as well as a clearer connection between historical policies and current challenges.

Methods:

The methods section is rigorous and follows PRISMA-ScR guidelines but requires further clarity. While grey literature broadens the scope, it reduces reproducibility unless rigor in its selection is explicitly addressed. Justify its inclusion by emphasizing its relevance to policy gaps and triangulation with peer-reviewed data. Additionally, the rationale for using the UHC framework should be articulated, and the choice of a scoping review should be justified as suitable for exploring broad evidence landscapes. These improvements will enhance transparency and accessibility.

Results:

The results section is well-structured and integrates multiple tables and figures to present data effectively. However, it relies heavily on narrative descriptions, often duplicating insights already presented visually. We recommend reducing narrative redundancy by focusing on key observations and integrating them into figure captions. Comparative benchmarks or annotations in visuals would highlight critical patterns and disparities. Expanding visuals for service and population coverage will provide a more balanced representation of UHC components.

Discussion:

The discussion section provides a strong analysis of Nepal’s health financing gaps and aligns well with the UHC framework. However, the recommendations are often general or too bold and would benefit from more context-specific strategies tailored to Nepal’s challenges, such as addressing NHIP dropout rates or increasing fiscal space. Incorporating global benchmarks would situate Nepal’s progress relative to peers. The feasibility of recommendations and inclusion of emerging trends like digital health or climate resilience should be explored to enhance practicality and relevance.

Conclusion:

The conclusion summarizes the findings well but lacks specificity in its recommendations. Including actionable steps tailored to Nepal’s context and prioritizing reforms based on feasibility and resource constraints would add value. A forward-looking perspective on emerging trends like digital health and climate-resilient systems would enhance relevance. Avoiding repetition of findings from earlier sections will make the conclusion more impactful.

References:

Several references are incomplete, lacking DOIs, URLs, or essential identifiers. For example, References 16 (Annual Health Report), 25–30 (Government Health Sector Strategies), 125–128 (journal articles and WHO reports), 130 (World Bank data), and 133 (Social Science & Medicine article) require additional details for traceability. The citation style resembles Vancouver but is inconsistently applied, especially for grey literature. We recommend reviewing all references for completeness, ensuring adherence to PLOS ONE guidelines, and including DOIs or publisher details where applicable to improve accuracy and accessibility.

Reviewer #3: The manuscript is well-written and provides a clear understanding of the subject matter. However, minor adjustments are recommended as follows:

1. In the abstract, consider shortening the background to allocate more space for summarizing the results and conclusions.

2. Although the methodology is robust, include additional details on how discrepancies among reviewers were resolved during data extraction to enhance the study's reproducibility.

3. Ensure explicit citation of data sources, particularly for figures and tables, to maintain transparency, and confirm that no data privacy issues arise, especially regarding sensitive budget allocation data.

4. Strengthen the conclusion by incorporating a call to action for stakeholders, outlining specific next steps for research and policy.

5. Conduct a final proofreading pass to address minor grammatical errors and typos for polished language.

**Do you want your identity to be public for this peer review?** For information about this choice, including consent withdrawal, please see our Privacy Policy

Reviewer #1: No

Reviewer #2: **Yes: ** Thamburaj Anthuvan

Reviewer #3: **Yes: ** Mohsin Hassan Alvi

---

## [Author Response · Author response to Decision Letter 1]

30 Jan 2025

Point by point response the reviewers’ feedback

Reviewer 1

Introduction

1. The health system of Nepal was well described in the introduction. However, the introduction lacks a description of how the epidemiology has changed in the country. Could you provide details of the changes in the disease burden the country has experienced? The 2019 burden of disease report may be helpful. https://nhrc.gov.np/wp-content/uploads/2022/02/BoD-Report-Book-includ-Cover-mail-6_compressed.pdf

Author’s response: Thank you so much for your insightful suggestions and for sharing the important report. We have added the description of how the epidemiology has changed in the country, as suggested.

2. Page 5, Lines 100-101 “Nepal is afflicted by a double burden of both high infectious and non-communicable diseases (NCDs).” This statement is a contradiction of what you stated in the abstract (transition to NCDs etc but not dual burden). Clarify, for consistency.

Author’s response: Thank you for your valuable suggestion. We have rephrased it as shifting/transitioned to NCDs in the revised manuscript.

3. The last two paragraphs in the introduction seem disjointed yet they are related. Including information on how the 2015 constitution of Nepal reflected the UHC ambitions and whether there are aspects of UHC that are not addressed by the constitution is likely to be beneficial to the reader.

Author’s response: Thank you for your valuable suggestions. The description of BHS is provided in the first paragraph, and the constitution guarantees BHS as a fundamental right. We have made the revisions as suggested in the revised manuscript.

4. The research gap that the study sought to address is not well elaborated in the introduction. You state that “Nonetheless, there is limited discussion about how health financing policies and strategies are in place to address these issues.” How limited is this discussion? Has there been any literature about the same? What financing policies are these, anyway? Have some studies reviewed some of these policies ready? And if yes, what did they find or what was the scope of the review? How different are they from this study? You should consider addressing these questions to give the reader a clear and well elaborated picture if the existing research gap that you are trying to fill with this study.

Author’s response: Thank you very much for your valuable comments. We have added more details to the last paragraph of the introduction section. We reviewed several policy documents, health insurance programs, and the challenges these programs face in achieving financial protection. Additionally, we have rephrased the statement on the research gap in the introduction section, as suggested.

5. In the last two sentences of the introduction, you mention what you have done in this study and the likely implication. Rephrase the sentences and clearly state the aim of your study reflecting the scope (the status and challenges of financing of health policies and programs towards universal health coverage).

Author’s response: Thank you for your valuable suggestions. We have rephrased the sentences to clearly reflect the aim of the study, including its scope, as suggested.

6. Ensure that the aim in the introduction and abstract are well aligned/the same.

Author’s response: Thank you for your comment. We have reviewed and ensured that the aim stated in the introduction is well aligned with that in the abstract.

Methods

7. The methods section is overly summarized which may have resulted into omission of relevant information for example in the search strategy: how were the search terms identified, were the search strings customised to the databases, was the search automated? What platforms were used if any? Concerning the selection of the articles: who did the selection, was it done by one or more people and how? What was the process for selecting the articles, was any platform used? Etc. Data extraction: how was the data extracted? Was it automated or done by individuals, did they use a template or program? For reproducibility, a more detailed methods section should be provided.

Author’s response: Thank you for bringing this to our attention and allowing us to clarify it in more detail. The search terms were identified through a literature review and expert consultations, and were agreed upon by all authors. The search terms used in this study included: “social risk protection” OR “health financing” OR “health insurance” AND “universal health coverage” OR “population coverage” OR “service coverage” OR “financial risk protection” OR “financial health coverage” AND Nepal*. We used three databases (PubMed, Embase, and Scopus) to identify academic literature, and also conducted hand searches of references from selected papers to locate additional studies not captured in the database searches. We also consulted the websites of relevant Ministries and their entities to access grey literature, including the latest reports, assessments, surveys, policies, plans, and health financing strategies. The literature search was initially conducted based on titles and abstracts by the first author and further assessed by the second author. We have added more details to the methods section in the revised manuscript.

8. The study states that PRISMA-ScR guidelines were followed. It is great that the checklist was attached. The structure of the methods section should be aligned with the requirements of PRISMA-ScR guideline. This will also enable you have a more detailed methods section making it easy to follow. This could be done by including subheadings reflecting the main components of the guideline.

Author’s response: Thank you for your valuable suggestions. We have included subheadings that reflect the main components of the guideline, as suggested, in the methods section of the revised manuscript.

9. Specify the reference for the WHO UHC framework used as there are a number of variations for example a recent one http://www.uhcwpr.info/. Please include a website as part of the reference. Page 6, lines 132-136.

Author’s response: Thank you for your suggestions. This webpage is not functioning properly at the moment. However, we have updated the reference to the WHO webpage, as advised.

10. Replace “and” with “or” in the statement “… in Nepali and English …” Page 6, line 137

Author’s response: We have replaced "and" with "or" in the statement, as suggested.

11. Clearly state the key words used in the search. Were “Health financing” AND "Universal Health Coverage” and “Nepal”?

Author’s response: We have provided more details about the key terms used in the search, which included: "social risk protection" OR "health financing" OR "health insurance" AND "universal health coverage" OR "population coverage" OR "service coverage" OR "financial risk protection" OR "financial health coverage" AND Nepal*.

12. State the date when the search was date.

Author’s response: The data search was conducted on the first week of May 2024. We have included the search date in the revised manuscript.

13. Include details of the strategy used to search for grey literature both online and offline (if done).

Author’s response: Thank you for your valuable comments. To identify the grey literature, we discussed it among the author team and consulted with our network working in the health financing space in Nepal. The first four authors have experience with the Nepalese health system and are familiar with both academic and grey literature on this topic. Additionally, we searched the web pages of various Ministries and their associated entities in Nepal. We also conducted searches using similar terms on Google and government websites. We have added further details on the strategy used to search for grey literature in the revised manuscript.

14. Attach as supplementary material, the search strings used in searching the databases or platforms used.

Author’s response: Thank you so much for your suggestion. We have attached the search strings used in search the database as supplementary material.

Results

15. Include a table of characteristics of the included articles, and records

Author’s response: We have included a table in the supplementary file and provided a summary of the table in the revised manuscript, as suggested.

16. You applied the WHO UHC framework (UHC cube), I would suggest that you reflect this in table 1 too by splitting both the status and Issues and challenges columns into three columns each to reflect the three dimensions of the UHC framework.

Author’s response: Thank you so much for your suggestions. We greatly appreciate your feedback. However, we believe that many of the issues and challenges are interconnected and overlap, particularly at the policy level. Therefore, we feel that the current formatting and presentation of the content in the table, at an aggregate level, is appropriate. We have further explained these issues and challenges in thematic form later in the manuscript, under each component of UHC. The presentation of the content in both summary form and component-specific narrative explanations complement each other and help reduce duplication of content.

17. Table 1: Some of the statements are general and could benefit from making them more specific. For example, when you state inequity, what aspect are you talking about, who are the disadvantaged, is it the poor, is it women? etc. When you mention poor public satisfaction with CBHI, what aspect of CBHI were people unhappy with, was it the service package, or something else? It will be helpful to be more specific in the Issues and challenges column.

Author’s response: Thank you so much for your important comments. We have included the specific information, as suggested, in Table 1 of the revised manuscript.

18. Table 1: I would have expected you to include the National Health Insurance Act 2017 as one of the major health financing policies on the list in table 1. Was there a reason of leaving it out?

Author’s response: Thank you for your suggestions. We have included the National Health Insurance Act 2017 on the list in table 1.

19. Table 2: Include a specific column for the name of the program that you are referring to in each row, separate from the actor(s) managing the program. This will make the table clearer and easier to follow.

Author’s response: we have revised as suggested.

20. Table 2: The scope of your study did not include functions but programs and policies. Please clarify why you are including health financing functions in the table.

Author’s response: we have removed the health financing functions from table 2 to keep content within the scope of this review.

21. Some of the sections provide explanation of the results. The explanation should be moved to the discussion. Page 20, Lines 160 to 171. Please cross check pages 15 to 20 to exclude the possible explanations provided and move them to the discussion section.

Author’s response: Thank you so much for pointing this out. We have moved it to the discussion as suggested.

22. The results section seems to have references that were not part of the articles included in the study for example reference number 97 (IQAir. Air quality in Nepal Air quality in Nepal 2022). Such references and the respective statements they support should be moved and considered for the discussion section.

Author’s response: Thank you so much for bringing this into our attention. We have revised it as suggested in the revised manuscript.

Discussion

23. The discussion explains the results to a great extent but misses comparing the findings in this study with similar studies that were done in Nepal or comparable settings. It will be great to include this comparison in the discussion.

Author’s response: Thank you so much for your important comments. We have included similar studies conducted in Nepal or in comparable settings, as suggested, in the revised manuscript.

24. One of the intentions of your study is to see whether there has been a mismatch between the health financing policies and the shifting demographics and disease burden in Nepal. However, this has not been discussed much and neither commented about in the first paragraph of discussion. Consider including a sub section on whether the policies are responding to the demographic and epidemiological shift and add a comment about the same in the first paragraph of the discussion section.

Author’s response: Thank you so much for your insightful comments. We have revised it as suggested in the revised manuscript.

25. The sub section entitled “Policy and plans versus translation into programs and services” has content that does not fully reflect the title. The current content has elements beyond the scope of the title (page 22, lines 207-221) and omits other relevant content, it would be great to provide more relevant examples. Some of the content especially regarding the details on financial protection my fit better in the subsections that follow.

Author’s response: Thank you so much for your suggestions. We have updated it as suggested to reflect the content as per the title in the revised manuscript.

26. Some of the subsections should be considered for merging as they discuss a similar area for example “Health insurance program in limbo”, and “High OOP and catastrophic expenditure”.

Author’s response: Thank you for your suggestions. We have revised and merged the two subsections, as suggested, in the revised manuscript.

Strength and limitations

27. Not interviewing people is part of the design of a scoping review in most cases so there is no need to mention it.

Author’s response: Thank you so much. We have revised the manuscript accordingly.

Reviewer #2:

Title:

The current title is informative but overly lengthy, which may impact readability. We suggest revising it to "Health Financing Amid Nepal's Demographic Shifts: A Scoping Review Using the Universal Health Coverage Framework." This revision improves brevity, maintains clarity, and enhances reader engagement while retaining all critical elements and better aligning with the paper's scope.

Author’s response: Thank you so much for your insightful suggestions. We have revised it as suggested.

Abstract:

The abstract is overly lengthy, with excessive background detail overshadowing the methodology, findings, and recommendations. We suggest condensing the background to focus on critical transitions (e.g., rising NCD burden, financial challenges) and aligning it more closely with the study's objectives. Prioritizing key results with quantifiable data (e.g., NHIP enrollment rates, budget allocation gaps) will improve clarity and impact. Recommendations should include actionable insights to demonstrate the study's practical contributions.

Author’s response: Thank you so much for your important suggestions. We have revised it to make it more concise, as suggested, in the revised manuscript.

Keywords:

The keywords are relevant but can be optimized for precision and searchability. Terms like "policy and strategy" are too broad and should be replaced with "health financing policies" to better capture the paper's content. Combining "service coverage" and "population coverage" into "universal health service and population coverage" would streamline the terminology. Adding "Nepal" as a keyword would improve geographic specificity and relevance.

Aurthor’s response: We have included the keywords in the revised manuscript as suggested.

Introduction:

The introduction provides essential context but is overly detailed, with outdated references that reduce its relevance to current challenges. We recommend replacing older references with recent studies and reports (from the past 5 years) on NHIP, UHC progress, and health financing trends. The section would benefit from a focused problem statement and earlier presentation of study objectives, as well as a clearer connection between historical policies and current challenges.

Author’s response: Thank you very much for your comments. We have included recent studies and revised introduction section as suggested in revised manuscript.

Methods:

The methods sectio

---

## [Decision Letter · Decision Letter 1]

4 Feb 2025

Dear Dr. Khatri,

Thank you for submitting your manuscript to PLOS ONE. After careful consideration, we feel that it has merit but does not fully meet PLOS ONE’s publication criteria as it currently stands. Therefore, we invite you to submit a revised version of the manuscript that addresses the points raised during the review process.

We look forward to receiving your revised manuscript.

Kind regards,

Himanshu Sekhar Rout, PhD

Academic Editor

PLOS ONE

Journal Requirements:

Reviewers' comments:

Reviewer's Responses to Questions

**Comments to the Author**

Reviewer #1: All comments have been addressed

Reviewer #2: (No Response)

Reviewer #3: All comments have been addressed

2. Is the manuscript technically sound, and do the data support the conclusions?

Reviewer #1: Yes

Reviewer #2: Partly

Reviewer #3: Yes

3. Has the statistical analysis been performed appropriately and rigorously?

Reviewer #1: N/A

Reviewer #2: Yes

Reviewer #3: Yes

4. Have the authors made all data underlying the findings in their manuscript fully available?

Reviewer #1: Yes

Reviewer #2: Yes

Reviewer #3: Yes

5. Is the manuscript presented in an intelligible fashion and written in standard English?

Reviewer #1: Yes

Reviewer #2: Yes

Reviewer #3: Yes

Reviewer #1: The authors have addressed all my comments well. I have no further comments. Good job. Looking forward to read the published version if accepted.

Reviewer #2: Thank you for your revisions. The manuscript has improved in structure and clarity, but a few areas still need refinement to strengthen its academic rigor and policy relevance. Below are specific suggestions for improvement:

1. Introduction

The introduction provides useful background, but the research gap is not clearly stated.

Consider adding a direct statement on what knowledge gap this study fills, particularly how it builds on or differs from existing health financing research in Nepal.

2. Methods

The PRISMA-ScR framework is well presented, but there is no mention of how data discrepancies were handled during extraction. Please include a short statement on how inconsistencies in data collection and classification were resolved to improve transparency and reproducibility.

3. Results

Some sections of the results are too narrative-heavy despite already having well-presented figures and tables.

Reduce excessive text where data is already visualized to improve readability.

4. Discussion

The discussion remains descriptive rather than analytical. Strengthen this section by discussing why certain health financing interventions have/have not worked in Nepal, linking findings to structural or policy constraints. A comparative benchmark would add value—how does Nepal’s health financing situation compare with similar LMICs in South Asia? (this was A more critical evaluation of government policies and health financing gaps is needed to make the discussion more insightful.

5. Conclusion & Policy Recommendations

The conclusion is still too general and does not clearly outline next steps. Please add 2-3 specific policy takeaways, such as recommended financing mechanisms, potential regulatory adjustments, or areas needing further research. Policymakers should be able to extract actionable insights directly from this section.

6. Figures & Exhibits

Some figures and exhibits do not add analytical value and could be moved to supplementary materials or removed.

Specifically, Figures 2, 4, and 6 can be safely removed or moved to supplementary material as they are redundant and do not directly support key arguments in the discussion.

Reviewer #3: I have no further comments for the authors. I believe, this manuscript is now ready for publication.

**Do you want your identity to be public for this peer review?** For information about this choice, including consent withdrawal, please see our Privacy Policy

Reviewer #1: No

Reviewer #2: No

Reviewer #3: **Yes: ** Mohsin Hassan Alvi

---

## [Author Response · Author response to Decision Letter 2]

20 Feb 2025

Point by point response to reviewer comments

Submission ID: [PONE-D-24-23054R1] - [EMID:dd6ac99af9acd14a]

Title: Navigating Nepal’s Health Financing System: A Path to Universal Health Coverage Amid Epidemiological and Demographic Transitions

Dear Editor and Reviewers,

We extend our heartfelt gratitude to you for diligently managing and evaluating our manuscript. Your insightful comments and valuable recommendations have significantly enhanced the quality of our work. In response to the reviewers’ feedback, we have meticulously revised the manuscript, addressed each comment and made the necessary modifications.

Additionally, we have attached a comprehensive response letter that outlines how we have incorporated the reviewers’ suggestions into the revised manuscript. Authors’ responses are italicized text for clarity.

Thank you once again for your time and effort in improving our manuscript.

…………………………………..

Sincerely,

Resham B. Khatri, PhD MPH MA

on behalf of all co-authors

Reviewer #1: The authors have addressed all my comments well. I have no further comments. Good job. Looking forward to reading the published version if accepted.

Author’s response: authors team would like to thank reviewer for this feedback, thank you so much.

Reviewer #2: Thank you for your revisions. The manuscript has improved in structure and clarity, but a few areas still need refinement to strengthen its academic rigor and policy relevance. Below are specific suggestions for improvement:

Author’s response: The authors team would like to thank reviewer for this feedback. Authors would like to appreciate reviewer for this opportunity to revise further our manuscript.

1. Introduction

The introduction provides useful background, but the research gap is not clearly stated.

Consider adding a direct statement on what knowledge gap this study fills, particularly how it builds on or differs from existing health financing research in Nepal.

Author’s response: we appreciate and thank reviewer for this comment. We further trimmed and focus the of this research. We have revised the introduction adding a short paragraph on knowledge gaps and significance of this study at end of the introduction section.

2. Methods

The PRISMA-ScR framework is well presented, but there is no mention of how data discrepancies were handled during extraction. Please include a short statement on how inconsistencies in data collection and classification were resolved to improve transparency and reproducibility.

Author’s response: We again thank reviewer for this feedback. We have added statement and described how data discrepancy was handled during extraction and analysis process. Refer to the “Extraction and charting of data” section in page 8.

3. Results

Some sections of the results are too narrative-heavy despite already having well-presented figures and tables. Reduce excessive text where data is already visualized to improve readability.

Author’s response: In agreement with the reviewer’s suggestions, we have significantly revised the results section to minimize the text that was already presented in tables and figures. This results table provides an opportunity for a robust discussion on how health financing functions should be structured in terms of risk pooling, payment mechanisms, and strategic purchasing.

4. Discussion

The discussion remains descriptive rather than analytical. Strengthen this section by discussing why certain health financing interventions have/have not worked in Nepal, linking findings to structural or policy constraints. A comparative benchmark would add value—how does Nepal’s health financing situation compare with similar LMICs in South Asia? (This was A more critical evaluation of government policies and health financing gaps is needed to make the discussion more insightful.

Author’s response: We greatly appreciate this feedback. Thank you, Reviewer. We have revised the discussion section to include some critique on the overall policy framing of Nepal’s health financing system (in line with constitutional, policy, and strategic development). This is followed by an analysis of existing health financing model (tax funded and premium based models), and then an analysis of policies and programs with respect to universal health coverage, strategies to strengthen the National Health Insurance Program (NHIP), and strategies to improve health financing functions in relation to revenue generation/ risk pooling, purchasing of services and payment mechanisms/resource allocation. We have also drawn lessons from health financing schemes in other countries with health system designs like Nepal. We believe the revised discussion section is more streamlined with the findings and aligns convincingly with the research question under inquiry.

5. Conclusion & Policy Recommendations- The conclusion is still too general and does not clearly outline next steps. Please add 2-3 specific policy takeaways, such as recommended financing mechanisms, potential regulatory adjustments, or areas needing further research. Policymakers should be able to extract actionable insights directly from this section.

Author’s response: thank you reviewer, we have revised final conclusions and policy implications as suggested.

6. Figures & Exhibits

Some figures and exhibits do not add analytical value and could be moved to supplementary materials or removed. Specifically, Figures 2, 4, and 6 can be safely removed or moved to supplementary material as they are redundant and do not directly support key arguments in the discussion.

Author’s response: We have moved figures 2,4, and 6 as supplementary file.

Reviewer #3: I have no further comments for the authors. I believe, this manuscript is now ready for publication.

Authors response. Author’s response: authors team would like to thank reviewer for this feedback, thank you so much.

Journal Requirements:

Please review your reference list to ensure that it is complete and correct. If you have cited papers that have been retracted, please include the rationale for doing so in the manuscript text or remove these references and replace them with relevant current references. Any changes to the reference list should be mentioned in the rebuttal letter that accompanies your revised manuscript. If you need to cite a retracted article, indicate the article’s retracted status in the References list and include a citation and full reference for the retraction notice.

Author’s response: authors team would like to thank editor for this feedback, we have rechecked references list again and ensure the journal requirement.

---

## [Decision Letter · Decision Letter 2]

11 Mar 2025

Dear Dr. Khatri,

Thank you for submitting your manuscript to PLOS ONE. After careful consideration, we feel that it has merit but does not fully meet PLOS ONE’s publication criteria as it currently stands. Therefore, we invite you to submit a revised version of the manuscript that addresses the points raised during the review process.

**Minor Revision** . 

We look forward to receiving your revised manuscript.

Kind regards,

Himanshu Sekhar Rout, PhD

Academic Editor

PLOS ONE

Journal Requirements:

Reviewers' comments:

Reviewer's Responses to Questions

**Comments to the Author**

Reviewer #2: All comments have been addressed

2. Is the manuscript technically sound, and do the data support the conclusions?

Reviewer #2: Partly

3. Has the statistical analysis been performed appropriately and rigorously?

Reviewer #2: I Don't Know

4. Have the authors made all data underlying the findings in their manuscript fully available?

Reviewer #2: Yes

5. Is the manuscript presented in an intelligible fashion and written in standard English?

Reviewer #2: Yes

Reviewer #2: Thank you for incorporating the revisions. The manuscript has improved significantly in structure and clarity. Below are a few minor refinements that would further enhance the paper’s academic rigor and impact:

1. Abstract

The abstract is now more structured, but it still lacks specific quantitative insights (e.g., exact figures on health financing gaps, enrollment rates). Consider incorporating key data points to strengthen the findings.

The contribution of the study could be highlighted more explicitly to make it clearer why this study is valuable.

2. Introduction

The introduction is better streamlined, but the research gap statement could be made clearer. A direct sentence explaining what this study adds beyond existing literature would improve clarity.

3. Methods

The PRISMA-ScR framework is well articulated, and the addition of a statement on data discrepancies is useful.

A brief mention of how grey literature bias was managed would further enhance the methodology section.

4. Results

The results section is much improved, with better clarity and reduced redundancy.

However, comparative benchmarks with other LMICs remain somewhat limited. Consider adding a brief comparison to place Nepal’s health financing in a broader global context.

5. Discussion

The discussion still leans towards descriptive rather than analytical in some areas.

A stronger evaluation of why certain interventions have or have not worked in Nepal would add more depth.

The comparisons with other LMICs have improved, but linking them more explicitly to Nepal’s policy context would further strengthen the analysis.

6. Conclusion & Policy Recommendations

The conclusion remains somewhat general and could be more action-oriented.

Please consider explicitly outlining 2-3 key policy recommendations that emerge from the study.

A short mention of next steps (e.g., future research areas, practical policy implementations) would add value.

7. Figures & Exhibits

Figures 2, 4, and 6 have been moved to supplementary materials as suggested—this is a positive change.

Figure 10 is still included, but its necessity should be reconsidered. If it does not provide critical analysis, it might be more suitable for supplementary materials.

Final Suggestion

With these minor refinements, the manuscript will be well-positioned for acceptance. These improvements will ensure that the study contributes meaningful insights to the discussion on health financing in Nepal. Let me know if further clarifications are needed.

**Do you want your identity to be public for this peer review?** For information about this choice, including consent withdrawal, please see our Privacy Policy

Reviewer #2: No

---

## [Author Response · Author response to Decision Letter 3]

17 Mar 2025

Point by point response to reviewer comments

Submission ID: [PONE-D-24-23054R2] - [EMID:dd6ac99af9acd14a]

Title: Navigating Nepal’s Health Financing System: A Path to Universal Health Coverage Amid Epidemiological and Demographic Transitions

Dear Editor and Reviewer,

We would like to thank you for reviewing our manuscript and providing us important feedback that has helped to improve the quality of our work. In this revised submission, we have tried our best to address the reviewer’s suggestions.

Additionally, we have attached a comprehensive response letter that outlines how we have incorporated the reviewers’ suggestions into the revised manuscript.

Thank you once again for your time and effort in providing constructive feedback on our manuscript.

…………………………………..

Sincerely,

Resham B. Khatri, PhD MPH MA

on behalf of all co-authors

Reviewer #2: Thank you for incorporating the revisions. The manuscript has improved significantly in structure and clarity. Below are a few minor refinements that would further enhance the paper’s academic rigor and impact:

1. Abstract

The abstract is now more structured, but it still lacks specific quantitative insights (e.g., exact figures on health financing gaps, enrollment rates). Consider incorporating key data points to strengthen the findings. The contribution of the study could be highlighted more explicitly to make it clearer why this study is valuable.

Thank you for highlighting the need to incorporate key data points in the findings section of the abstract. The updated abstract has now data points related to health financing situation of Nepal including out-of-pocket expenditure. Furthermore, the conclusion section in the abstract is refined to align with the changes made in the conclusion section of this paper.

2. Introduction

The introduction is better streamlined, but the research gap statement could be made clearer. A direct sentence explaining what this study adds beyond existing literature would improve clarity.

We now have improved this section by clearly spelling out the research gap and have highlighted the contribution of the study to the existing literature in the last paragraph of the introduction section

3. Methods

The PRISMA-ScR framework is well articulated, and the addition of a statement on data discrepancies is useful.

A brief mention of how grey literature bias was managed would further enhance the methodology section.

We tried to address data discrepancies by including figures from relevant authorities and incorporating latest figures. We also validated data and grey literature information from people who are closely acquainted with Nepal’s health financing system. A short narrative of how grey literature bias was managed has been incorporated in the “Selection of documents” sub-heading under the “Methods” section.

4. Results

The results section is much improved, with better clarity and reduced redundancy.

However, comparative benchmarks with other LMICs remain somewhat limited. Consider adding a brief comparison to place Nepal’s health financing in a broader global context.

We appreciate the reviewer’s suggestion to include comparative benchmarks with other LMICs in the results section. However, the authors believe that instead of highlighting these comparisons in the results, which primarily focus on Nepal’s health financing context, we have incorporated them in the introduction. In the introduction, we provide a brief comparison of Nepal’s health financing position within the broader global context, contrasting it with global and regional averages, as well as with data from other LMICs.

5. Discussion

The discussion still leans towards descriptive rather than analytical in some areas.

A stronger evaluation of why certain interventions have or have not worked in Nepal would add more depth.

The comparisons with other LMICs have improved, but linking them more explicitly to Nepal’s policy context would further strengthen the analysis.

Thank you for your feedback. We have refined the discussion section by reorganizing the subheadings and removing redundant or descriptive text. To further strengthen the argument and provide context and recommendations for Nepal’s health financing, we have included additional lessons and examples from other LMICs. Additionally, based on the findings and discussion of the study, we summarized and visually presented the implications for policy, program, and research in a figure. We believe these revisions have enhanced the discussion, as suggested.

6. Conclusion & Policy Recommendations

The conclusion remains somewhat general and could be more action-oriented.

Please consider explicitly outlining 2-3 key policy recommendations that emerge from the study.

A short mention of next steps (e.g., future research areas, practical policy implementations) would add value.

In response to the reviewer’s suggestions, we have reframed the conclusion to clearly summarize the key findings. Additionally, we have outlined four action-oriented policy recommendations aimed at addressing the existing gaps in health financing performance. We have also highlighted potential areas for future research that may be of interest to research stakeholders.

7. Figures & Exhibits

Figures 2, 4, and 6 have been moved to supplementary materials as suggested—this is a positive change.

Figure 10 is still included, but its necessity should be reconsidered. If it does not provide critical analysis, it might be more suitable for supplementary materials.

We have well-noted the feedback and have moved the figure 7 (figure 10 in previous track change version) in the supplementary section.

Final Suggestion

With these minor refinements, the manuscript will be well-positioned for acceptance. These improvements will ensure that the study contributes meaningful insights to the discussion on health financing in Nepal. Let me know if further clarifications are needed.

Thank you for this observation. We have tried our best to incorporate the feedback and we are hopeful that the study if published would be helpful to initiate discussion on health financing reforms in Nepal.

---

## [Editor Report · Decision Letter 3]

2 May 2025

Navigating Nepal’s Health Financing System: A Path to Universal Health Coverage Amid Epidemiological and Demographic Transitions

PONE-D-24-23054R3

Dear Dr. Khatri,

We’re pleased to inform you that your manuscript has been judged scientifically suitable for publication and will be formally accepted for publication once it meets all outstanding technical requirements.

Kind regards,

Himanshu Sekhar Rout, PhD

Academic Editor

PLOS ONE

Additional Editor Comments (optional):

Thank you for incorporating all points raised by the Reviewer 2. Now the paper is accepted for publication
---

## [Editor Report · Acceptance letter]

PONE-D-24-23054R3

PLOS ONE

Dear Dr. Khatri,

I'm pleased to inform you that your manuscript has been deemed suitable for publication in PLOS ONE. Congratulations! Your manuscript is now being handed over to our production team.

Kind regards,

on behalf of

Professor Himanshu Sekhar Rout

Academic Editor

PLOS ONE